# Internet of Things: Security and Solutions Survey

**DOI:** 10.3390/s22197433

**Published:** 2022-09-30

**Authors:** Pintu Kumar Sadhu, Venkata P. Yanambaka, Ahmed Abdelgawad

**Affiliations:** 1College of Science and Engineering, Central Michigan University, Mount Pleasant, MI 48858, USA; 2Department of Mathematics and Computer Science, Texas Woman’s University, Denton, TX 76204, USA

**Keywords:** Internet of Things, security and privacy, cryptography, physical unclonable function, blockchain, authentication framework

## Abstract

The overwhelming acceptance and growing need for Internet of Things (IoT) products in each aspect of everyday living is creating a promising prospect for the involvement of humans, data, and procedures. The vast areas create opportunities from home to industry to make an automated lifecycle. Human life is involved in enormous applications such as intelligent transportation, intelligent healthcare, smart grid, smart city, etc. A thriving surface is created that can affect society, the economy, the environment, politics, and health through diverse security threats. Generally, IoT devices are susceptible to security breaches, and the development of industrial systems could pose devastating security vulnerabilities. To build a reliable security shield, the challenges encountered must be embraced. Therefore, this survey paper is primarily aimed to assist researchers by classifying attacks/vulnerabilities based on objects. The method of attacks and relevant countermeasures are provided for each kind of attack in this work. Case studies of the most important applications of the IoT are highlighted concerning security solutions. The survey of security solutions is not limited to traditional secret key-based cryptographic solutions, moreover physical unclonable functions (PUF)-based solutions and blockchain are illustrated. The pros and cons of each security solution are also discussed here. Furthermore, challenges and recommendations are presented in this work.

## 1. Introduction

Devices like the Internet of Things (IoT) have a significant presence in our daily lives owing to the technological revolution, wireless devices, and communication systems. IoT has become a vital part of the digital era of Industry 4.0. The ability to bring physical things into the digital world is becoming more likely because of technology [1]. IoT networks have an impact on a variety of areas, including home monitoring and daily life patient monitoring. IoT integrates the advantage of data processing, analytics, and draws out the power of the web to make decisions for physical objects of the real world. It is a system where intelligent objects are linked and access the internet as the foundation of the interconnection to gather and share information using “Things”. IoT has become one of the main focuses of research across the world.

### 1.1. Paper Organization

To depict the overall scenario and solutions, this paper is organized as Figure 1. To give an overview of the IoT ecosystem, security threats, and respective solutions to the new and existing researchers, this paper does a comprehensive review. To provide a better view, this paper shows the various applications of IoT, different security threats, security mechanisms, authentication frameworks, and future directions sequentially. Section 1 presents the overview of IoT, IoE, and data security and privacy. Section 2 shows the related work. The overview of different IoT applications, their importance, prospects, functionality, impact, and challenges are discussed in Section 3. After that, security constraints of IoT devices, attacks on the TCP/IP reference model’s various layers, and attacks on each layer of the IoT network in detail are discussed in Section 4. Different cryptographic operations are presented in Section 5. Moreover, existing cryptographic security solutions with pros and cons are discussed in Section 6. Furthermore, this paper shows security challenges and future directions in Section 7.

Table 1 presents the notations that are commonly used in the paper.

### 1.2. Motivations

IoT seeks to link equipment to the web to make it accessible anytime, anywhere, and by anyone. With the help of seamless connectivity and smart objects like washing machines, microwave ovens, meters, vehicles, mobile phones, refrigerators, medical devices, etc., IoT is making remarkable applications such as intelligent transport systems, smart healthcare, smart homes, smart cities, etc. Ericsson forecasted that by 2022 there would be 29 billion connected devices, among which 18 billion will be IoT devices. Considering the vast utilization of IoT devices, the application of IoT and security requirements have increased. Moreover, many devices will be placed in rural areas which will be unattended. An attacker can compromise the devices and find an entry point to compromise the network. Research interest in IoT areas is increased a lot and it has become one of the first priorities among industry and academia, which can be found in the existing literature. Many survey papers show a glimpse or partial of IoT security threats or solutions. We have reviewed many existing studies related to IoT applications, different attacks, and security mechanisms. This paper aims to show security vulnerabilities in IoT devices as well as IoT networks with security solutions to provide insights to the network designers and motivate future research directions for both new and present researchers. The motivation of the paper is to provide all relevant information from existing literature under an umbrella that can benefit future readers by getting the required information from an article.

### 1.3. Internet of Things

The term ‘IoT’ is also termed as ‘Internet of Objects’. IoT devices are of varying sizes and capabilities of electrical or electronic devices that are capable of connecting to the web. IoT devices can be utilized in a variety of settings, including those shown in Figure 2, including residence, manufacturing, environment, healthcare, electricity, and communication.

IoT can be defined from several perspectives in the literature. Figure 3 shows the elements and paradigm of IoT [1,2]. In the case of the things-oriented vision, the aim of IoT is to become smart through the collaboration and focus of both virtual and physical bodies. These devices see, hear, think, share information, and perform tasks by coordinating decisions [2]. The Internet-oriented vision focuses on the development of IP-based networks so that things can connect and communicate with each other [3]. In the IoT systems, the demand of high volume data flows from and to sensors or smart entities emerges semantic-oriented visions [4]. In the IoT system based on service-oriented vision, IoT intelligent services and applications based on the three perspectives mentioned above are concentrated [5].

Six main elements are required to deliver the functionality of IoT, as shown in Figure 3 [2]. Among six elements, identification is crucial for naming and matching services with their demand. IoT devices capture data by sensing and send data to the cloud/database for analysis. The element of communication is used to bind heterogeneous objects simultaneously to serve specific digital services. WiFi [6], Bluetooth [7], Zigbee [6,8], MQTT [9], IEEE 802.15.4, OPC-UA, NFC, Z-wave, LoRaWAN [10], SigFox, and LTE-Advanced are a few examples of communication protocols. Hardware elements such as microcontrollers, microprocessors, system-on-chip (SoCs), and Field-programmable gate array (FPGAs) are used for processing. Processing units and software applications are used for computing. The combination of both hardware and software elements represents the brain of IoT. The ultimate goal of IoT is to deliver the services at such a level so that it is capable of providing services anytime, anywhere, to anyone.

Generalized IoT system architecture has four layers which are Perception, Network, Processing, and Application layer which is shown in Figure 4 [11,12]. The devices in the perception layer such as sensors of different types, Radio-frequency identification (RFID) scanners, surveillance cameras, Global Positioning System (GPS) modules, conveyor systems, industrial robots, etc., are responsible for monitoring conditions, collecting sensory data, etc. Different communication systems like WiFi, Bluetooth, Zigbee, LTE, etc., with protocols like IPv4, and IPv6 consist of the network layer responsible for data transfer to the next layer’s processing system [13]. Typically, cloud servers and databases in the processing layer are responsible for data analyzing, computing, decision making, and storing a huge amount of data. As per the requirement of users, the application layer delivers the specific needs of the end-users.

### 1.4. Internet of Everything

IoT is considered the backbone of many smart applications such as the manufacturing industry, smart healthcare, intelligent transportation, smart grid, smart city, etc. “Instrumentation”, “Interconnections”, and “Intelligence” are essential elements of smart applications of IoT and also referred to as 3Is. On the other hand, IoT acts as an integrated component of IoE [14]. Initially, Cisco coined the concept of IoE in 2013 [15]. The primary objective of the IoE technology is to transform gathered data into information or actions, and assist data-based relevant decision-making. Moreover, IoE aims to facilitate new capabilities, proficiency, and great experiences to become a self-contained and productive system. Figure 5 presents the main “four pillars” of IoE [16,17,18]: (1) People, (2) Data, (3) Process, and (4) Things.

*People*, in a system, are a critical element of the IoE environment. With the introduction of IoE, people share their personal insights through innumerable new ways of communication such as social networks, data-collecting smart sensors, actuators, smartwatches, etc. These data are being transmitted to the servers for analyzing and providing relevant information according to their personal, system, industry, or business demands. The information assists the people or system in quickly resolving open issues or reach to decisions.*Data* are transferred as a traditional IoT network. Data, complete or partial, collected by devices could either be sent directly or after initial transformation in the edge layer. The raw data captured or generated by the device has no importance. Nevertheless when these pieces of data are transformed, summarized, classified, and analyzed by the device itself or by the cloud server at the edge layer, it becomes priceless content that can monitor and control numerous systems, make accurate and faster decisions, and entitle smart solutions.*Process* based on various systems such as artificial intelligence, deep learning, social networks, computer vision, or other technologies helps to deliver the proper information to the designated people/device/place at the expected time. Through this process, information will be extracted from data, and the data communication will be controlled through the network. The purpose of processes is to get the optimum outcome for further processing or decision making.*Things* encounter the definition of IoT. Different kinds of sensing elements are embedded with physical items that serve the purpose of the data collection. Different devices must have communication capabilities, wireless or wired, for transmitting generated and processed data to the right destination across the system.

Figure 6 shows the general architecture of the IoE system [15,19]. The IoE system is a combination of blocks of a visualized data center, intelligent network, and connected devices. A virtual data center consists of an operating system, desktop virtualization software, etc. It communicates with an intelligent network to provide services to connected devices (smart sensors, devices, actuators, mobile terminals, wearable devices, etc.) and ultimately human beings. Connected devices are categorized into three segments: human to human, machine to machine, and machine to human. The backbone is an optical fiber network to guarantee a high velocity of the networks to ensure a high presence, low latency, and excellent quality of IoE services [15]. The fiber-optic network can also be replaced with a wireless network.

### 1.5. Data Privacy and Security

Secure communication is the prime factor for maintaining data privacy and confidentiality in a different kind of IoT and IoE architecture. Regulations linked to data collection, storing in memory, and exchanging details should follow a way that can preserve the user’s personal record is referred to as data privacy. Security can be improved using secure key management [20,21] and physical unclonable function [22]. Figure 7 shows four domains that defines various security concept of IoT-based network [23].

As IoT is becoming an integrated part of our daily life, the usage of IoT-based devices is increasing rapidly. It is predicted that 70% of devices will be IoT-based devices due to the continuous development of urbanization. CISCO predicted that $14.4 trillion devices will be exploited by 2025 [16]. M2M traffic is increasing, and it is expected that it will be up to 45% of the whole Internet traffic by 2022. Another study shows that by 2025 global economy of IoT-based healthcare applications will contribute about $1.1–$2.5 trillion in growth per annum. It will change the global economy, and $2.7 trillion to $6.2 trillion is the estimated impact by 2025 [2]. By 2025, it is reckoned that 75 billion IoT devices will be present in the global network. In the worldwide network, the growth of IoT devices is attracting attackers to gain access to fulfilling their goals. According to Symantec, since the rapid expansion and development, cyber-criminals progressively target IoT technology. In 2019, cyber attacks increased by 300% compared to 2018, and it was approximately 3 billion attacks [23]. In [24], it is also stated that security is the major challenge of the IoE network. Laura DeNardis reckons the threats, and risks of IoE in *The Internet in Everything: Freedom and Security in a World With No Off Switch*. Privacy, cyber-physical security, and interoperability politics, economic growth, individual rights, business models, governance are also discussed in the book [25].

## 2. Related Work

IoT has a major impact on people’s daily life which attract researchers to contribute more so that it could become beneficial for human being. Consequently, many researchers worked on the survey of IoT to provide knowledge regarding the IoT ecosystem and its details. A few works were done to provide an overview of challenges faced by IoT. The security threats are reviewed in [11,26,27,28]. In these articles, different kinds of attacks were discussed. Article [29] showed the security flaws in Bluetooth and possible attacks on IoT using vulnerabilities of Bluetooth. Challenges of IoT are presented by others and a few of them are [30,31,32], etc. The article [31] also presented security guidelines and the impact of 5G on IoT systems was discussed in [33]. IoT architecture and layers were focused on in [34] and different protocols are discussed in [28,34,35]. Various applications of IoT were discussed in different papers such as [36] showing the impact of smart logistics in the industry. As IoT is a resource-constrain device, efficient and lightweight operations are required. To cover these things, refs [37,38,39] showed how edge computing can help to process IoT services like smart agriculture, smart logistics, etc. In relation to the above type of researches, it is necessary to ensure secure data transfer and defend against different security attacks. The authentication framework can be designed in both centralized methods and decentralized mechanisms. Blockchain based, decentralize method, solutions were presented in different review papers, for example [11,40,41,42], etc. Smart mobile IoT architecture along with different security mechanisms was presented in [28]. On the other hand, machine learning based intrusion detection solutions were demonstrated in [27,43,44]. However, none of them reviewed the IoT attacks taxonomy, attack surfaces, security mechanisms, secure data communication method, etc., as we did in this research. Table 2 summarizes the contributions of different review papers and mentions how the perspective of our paper is different from other survey papers.

In this survey, the security and privacy risks of IoT are demonstrated. The issues and obstacles of different applications are highlighted. The contributions of this paper are the followings.

Overview and elements of both IoT and IoE networks. The differences between these are discussed.Limitations and Vulnerabilities of IoT devices and network. The taxonomy of different layers is provided in detail.Countermeasures of each kind of attack are provided with reference.Available security measures and their application in the sector of IoT are analyzed.Open issues of IoT security systems and future directions are also discussed.

## 3. IoT Applications

Applications of IoT can be utilized in various ways to assist systems and businesses in simplifying, improving, automating, and controlling processes. IoT can also be used for delivering important data, activity performance, or even environmental factors that have to be monitored continuously and remotely. IoT applications can therefore help with the creation of new systems and business strategies, as well as provide businesses with the instant data they need to create products and services.

### 3.1. Smart City

A smart city is a technologically advanced metropolitan region that collects information using different electronic techniques, voice recognition technologies, and sensors. The data is utilized to successfully handle assets, services, and programs; in turn, the information is being used to function smoothly throughout the city. Data obtained from community members, equipment, structures, and assets are processed and analyzed to track and maintain road and transport infrastructures, energy plants, utilities, water system connectivity, waste management, preventing crime, data management, education institutions, libraries, healthcare facilities, and other community programs. A smart city is a collection of different sensors and equipment to monitor, report, and process to manage the resources of the infrastructure effectively. Using the information gathered from the wireless sensors, the system will learn and make decisions to provide beneficial outcomes to the people. Compared to the healthcare, water supplies, and environment surveillance in the current urban areas, a smart city will be able to connect the citizens and the required services in a better way [45,46]. It is necessary to protect both the residents’ privacy and the information system’s integrity. Sensitive information is being gathered by sensing devices, which makes it vulnerable to cyberattacks.

### 3.2. Internet of Medical Things (IoMT)

The integration of health features into IoT devices makes the environment an IoMT. With the advancement of technology, the usage of IoMT devices is increasing. Moreover, the COVID-19 situation limits face-to-face meeting between patients and doctors. The pandemic has created a new era of IoMT for providing treatment to patients [47]. IoMT is making a network of people and medical devices (wireless medical devices and implanted medical devices). It utilizes wireless communication (Bluetooth, WiFi, 3G, 4G, 5G, ZigBee, etc.) to exchange health data with medical facilities like doctors, hospitals, medical experts, etc., [48]. With the advancement of microelectronics, medical devices have become intelligent and can monitor and report physical conditions such as blood pressure, heartbeat, oxygen level, etc. The devices can be placed in the body in the form of watches, belts, shoes, clothes, necklaces, etc., [49]. Moreover, IoMT has become the most significant change among the development in the medical sector as it brings not only the aged people but also every aged sick people in continuous monitoring and treatment. Especially people suffer even after recovering from COVID-19, and IoMT provides immediate treatment if necessary. Many healthcare systems from all over the world adopt the IoMT system to provide treatment. However, nearly 50% of IoMT equipment is susceptible to attack, as per the 2020 CyberMDX research. IoMT network is distinct from other systems in that they have the potential to impact patients’ lives and raise privacy problems if their identities are divulged [50]. Maintaining security and privacy is the primary concern of the IoMT system. According to a research from cybersecurity firm Critical Insights, cybersecurity incidents reached an all-time peak in 2021, compromising a record quantity of patients’ personal health data. Healthcare attacks harmed 45 million people in 2021, rising from 34 million in 2020. As per the research, the number of breached data has tripled within only three years, up from 14 million in 2018 [51].

### 3.3. Smart Grid

A smart grid is an electrical system that contains several efficient and energy efficiency features such as infrastructure for intelligent metering, intelligent power panels, smart equipment, control system, alternative/renewable energy, etc. The term “smart grid” refers to a concept that encapsulates the entire power generation and distribution system in a single frame. It is an electricity system built on digital technology that uses two-way digital communication to supply electricity to consumers. To put it another way, a Smart Grid is a grid that makes the entire system smarter or cleaner. Clean energy is presently in high demand all around the world. In 2003, the first time the term “Smart Grid” was stated by Michael T. Burr in an article [52]. Smart grid technology allows real-time monitoring, coordination, and control of the electric energy grid via communication networks between physical components, resulting in more effective and economical grid management. The widespread availability of Internet connectivity in most houses has made the smart grid more viable to adopt. The smart grid consists of Supervisory Control And Data Acquisition (SCADA), Energy Management System, Grid Communication Systems, and Distributed Energy Resources (DERs). In a smart grid system, users’ data privacy and security are crucial and challenging issues. A cyber-physical exploit is a safety failure in cyberspace that negatively affects a CPS’s physical environment [53]. There have been a number of significant cyber-physical incidents have been reported in the sector in recent years. A computer worm called “Stuxnet” leveraged four zero-day flaws and cryptographically signed certificates to evade intrusion detection. It hit the Iranian nuclear fuel enrichment complex in June 2010 where the programmable logic controllers (PLC) of the SCADA system were the targets [54]. Three Ukrainian electric power distribution firms were hacked in a coordinated operation in December 2015. Thirty substations were blacked out for almost three hours, resulting in 225,000 consumers experiencing wide-area power outages. To frustrate claims of disruptions, a telephonic DoS attempt was conducted while authorized members’ virtual private network accounts were stolen using the Black-Energy3 virus [55]. At present different AI based techniques are used to detect and defend the mechanism of the smart grid security system.

### 3.4. Internet of Vehicles (IoV)

With the development of industry, the number of vehicles is increasing rapidly. The increment of vehicles raises security concerns that trigger secure communication. IoV will lead industry 4.0. It is certain that the IoV will be bright and profitable in the future, offering improved road safety, reduced environmental effects, better space utilization, and cost control. The IoV, often known as smart transportation or connected cars, is a framework comprising vehicles, smartphones and wearables, roadside equipment, and a network. People, cars, and numerous IoT devices that are part of the transport system communicate over IoV [56]. Transportation, production, energy, software, and other industries are all affected. The IoV ecosystem includes hardware, software, services, and multiple network technologies ranging from Bluetooth and cellular to Wi-Fi and 5G, as well as several types of communication (V2V, V2X, and so on). Vehicle-to-vehicle and vehicle-to-infrastructure communication systems are combined to create the vehicular ad hoc network or VANET, and the term IoV has developed from the notation VANET. The combination of functionalities such as sensing units, control platforms, and various computer resources makes each vehicle in IoV an intelligent object. Each vehicle connects to any entity via a V2X communication architecture. The aim of IoV, also known as V2X, is safe driving by reducing accidents, alleviating traffic congestion, providing low traffic route information, and providing other information services. Every vehicle in the IoV network interacts with all other things that could have an influence on it. V2X mainly includes vehicle-to-vehicle (V2V), vehicle-to-sensors (V2S), vehicle-to-infrastructure (V2I), vehicle-to-network (V2N), and vehicle-to-pedestrian (V2P) communication. Nevertheless, roads can be seized by modifying or changing data or making wrong decisions due to the receipt of prank data [57]. To avoid these kinds of situations, it is required to develop a robust authentication framework that can resist security vulnerabilities and can conduct verification within milliseconds.

## 4. Devices’ Vulnerabilities and Requirements

Connected devices are suffering from different kind of threats, and it is increasing day by day. The low power devices can not satisfy the requirements of traditional methods to ensure safety. To preserve the security of the devices and the privacy of the consumers, it is essential to block the entry of adversaries into the devices or the network. In this section, the possibility of security threats and countermeasures will be illustrated.

### 4.1. Security Constraints

Applying the conventional security mechanics in IoT networks or devices is not straightforward as these devices are usually resource-constrained. The crucial security limitations of the IoT appliances are mentioned in Table 3 [58]:

### 4.2. IoT Attack Surfaces

With the growing number and variety of IoT devices, the attack surface multiplies many times. Furthermore, the attack surface of IoT networks is increasing with the raised population (number of IoT devices), convolution, heterogeneity, diversity, interoperability, portability, mobility, location, topology, and distribution of objects (devices, controller, connectivity, consumer, and services). The attack surface increases significantly as IoT device diversity and number increase. Additionally, the increased population (quantity of IoT devices), convolution, heterogeneity, diversity, interoperability, portability, mobility, location, topology, and dispersion of objects are all contributing to an increase in the attack surface of IoT networks (devices, controller, connectivity, consumer, and services). Enablers (such as networks and protocols) and entities make up an attack surface (i.e., devices, methods, and information). The attack surface is determined by the connectivity of a system’s components as well as policies that manage device permission for system access. IoT devices make several building blocks in the network of an IoT architecture. It is required to keep in mind different kinds of attack surfaces [59]. Administrative interface, device/cloud web interface, update mechanisms, mobile applications, physical interfaces, device firmware, device memory, etc., could be possible attack surfaces. An attack surface groups numerous locations an attacker can exploit to get access to a system and steal/leak/alter information. Behind each attack surface, there are particular elements and functions of devices of an IoT network where a set of security flaws lies. After identifying the attack surface, it is possible to identify security risks and potentially vulnerable areas where deep level protection is required. It is evident that the security of the IoT ecosystem is in the blink from various perspectives. The sheer amount of attack surfaces an attacker could use to carry out their harmful operations is undoubtedly a motivator to develop effective security solutions. Furthermore, due to the resource-constrained nature of IoT nodes, conventional security measures are unable to be implemented, putting the entire network in danger. The Mirai botnet and its derivatives, which can take control of IoT devices and launch a devastating DDoS attack, are excellent examples of such risks.

### 4.3. IoT Vulnerabilities

IoT devices are creating great experiences for consumers, and are everywhere around us. With the growth of IoT devices, security threats are also increasing as hackers are getting a chance to manipulate a huge amount of data in the vastly connected world. Without proper security in place, IoT devices will be vulnerable to sensitive data leakage. IoT and cybercriminal activity are invisible to the naked eye, and they can reach us anytime. Furthermore, IoT devices are vulnerable to assaults and security risks since they lack the essential built-in security to fight threats due to their low price, minimal power, and poor computing capability, as well as the network’s heterogeneity and scale. IoT devices are vulnerable to threats not only for technical aspects but also to users’ activity. Here are a few of the reasons why these smart devices are still at risk [60]:Limited computing capabilities and hardware limitations: These devices are designed for particular applications that require only limited processing capabilities, leaving minimal area for security and data protection to be integrated.Heterogeneous transmission technology: These devices communicate with different kinds of devices and frequently use various communication technologies, which make it difficult to establish uniform protection measures and protocols.Components of the device are vulnerable: Millions of smart devices can be harmed by insecure or outdated fundamental elements.Users lacking security awareness: Due to a lack of user security knowledge, smart devices may be exposed to risk zones and attack possibilities. Many IoT devices allow users to integrate third-party apps, which could also drive the device into a danger zone.Weak Physical Security: Unlike data centers of internet services, not only just users, but also others with heinous intents, have physical access to a major share of IoT components.

In [61], the security issues are divided into Software level threats, and Hardware level threats, as shown in Figure 8. Hacking, information leakage, illegal access, and more. are defined as software level attacks to force the system to malfunction and collect desired information like credit card information, password, etc. A firewall, updated virus database, and up-to-date software usage can limit software-based attacks. Not only software-level attacks but also hardware-level attacks are prominent scope for attackers. To build complete secure hardware, secure Integrated Circuits (IC) or SoCs are required to develop. It is becoming complex due to nano-scale design, distribution of embedded Very Large Scale Integration (VLSI) chip fabrication, and also the incorporation of third-party Intellectual Property (IP) core. Insertion of a single malicious circuit during the fabrication process can make the system compromised, and this can be invisible to designers.

In Figure 9, the pyramid of threat factors is represented based on vulnerabilities and impacts [23,62]. The top elements of the pyramid are more vulnerable with minimum probable impact and lower-level elements show opposite characteristics. It can be asserted that Cyber threats seek greater control and opportunities further down the stack.

There are defects in a system’s component that render it vulnerable and expand the attack surface. In particular, an adversary tends to exploit the hardware or software of the IoT system to gain access to perform their malicious activity. In the report of HP, they found that 50% of the commercially available IoT has a significant security flaw [58]. It is essential to prevent and react against previously listed vulnerabilities as they could expose sensitive information and exploit the IoT system. Since IoT network is exposed to different kinds of attacks, it is a complex task for the security analysis and imposes full-proof security measures. Nevertheless, the massive volume of data generated by IoT environments is leveraging the enhancement of the security level of the entire system.

### 4.4. IoT Security Requirements

It is necessary to understand the security objectives to secure equipment. Confidentiality, integrity, and availability are known as the CIA triad as traditional security targets in the state-of-the-art. Confidentiality is linked to a set of rules that establish criteria for authorized entities who have access to information. Integrity is another characteristic that ensures reliable services so that only legitimate commands and information are being received by IoT devices. Availability ensures that IoT functionalities are accessible by legitimate objects and users anytime and anywhere. IAS-octave, alluded to the Information Assurance and Security, is a comprehensive set of security goals that eliminates the drawbacks of the CIA triad [63]. Table 4 lists the IAS-suggested octave’s security goals, as well as their meanings and abbreviations [64].

### 4.5. IoT Attacks

IoTs are subjected to various extensive threats and security vulnerabilities which raises the necessity to establish security frameworks in different domains, including identification/authentication, confidentiality, reliability, and non-renunciation. We will talk about security attacks based on distinct properties in this section.

#### 4.5.1. Spoofing Attacks

A spoofing attack occurs when an attacker gains access to another device or user on a network. The adversary exploits the user’s device in addition to launching attacks against network hosts, stealing data, spreading malware, or getting around access controls [65]. In [66], spoofing attacks are distinguished into two types: (i) spoofing in the link layer: the whole communication between the two entities can be spoofed, and (ii) spoofing at the end to end layer: the attacker can counterfeit a certain service.

There are different types of spoofing attacks that can be initiated by attackers. Among different types of spoofing attacks, IP address spoofing attacks, ARP spoofing attacks, and DNS server spoofing attacks are some of the most frequent methods discussed here [67].

IP Address Spoofing Attacks: While a node communicates, it exchanges network data packets, each of which contains several headers for routing and transmission continuity. The ’Source IP Address’ header, for example, shows the packet’s sender’s IP address. In an IP address spoofing attack, an attacker falsifies the content of the origin IP header generally with random numbers in order to disguise itself. Most DDoS malware kits and attack scripts include IP spoofing as a standard feature. In two processes, IP spoofing attacks can make use of network traffic to flood targets. In one process, multiple spoofed addresses can be used to flood a selected target. This approach operates by sending more data to a target than it can manage. In another method, the victim’s IP address is spoofed, and packets are being sent from that address to numerous receivers on the network. When other nodes receive the packet, those will send a response to the target’s address, and the target will be flooded. Lack of controls against Denial of Service: Services can be targeted in such a way that they deny service to the entire network or device itself. The surface attack area is device network services for this case [68].ARP Spoofing Attacks: ARP (Address Resolution Protocol) is a protocol that translates IP to MAC and vice versa, allowing network communications to reach a particular destination on the network. ARP Spoofing Attacks, also called as ARP Poisoning, involve an attacker sending forged ARP answers over a neighborhood region conducive to getting the IP address of a legitimate member of the network, which will be used to link with the attacker’s MAC address. After connecting the attacker’s MAC address to a valid IP address, the intruder will start to receive any messages destined for that IP address. An attacker can use this technique to intercept, manipulate, or even interrupt data in transit, allowing them to carry out other types of exploits like denial-of-service, eavesdropping, session hijacking, and man-in-the-middle attacks. Only local area networks that use the Address Resolution Protocol are vulnerable to ARP spoofing attacks [69].DNS Server Spoofing Attacks: DNS stands for Domain Name System, and it is a system that converts website addresses, email addresses, and other human-readable domain names into IP addresses. In this attack, the attacker introduces corruption into the DNS resolver’s cache, which is utilized to redirect a given domain name to a different address, in this attack. The attacker’s server will be at the new IP address, including malware-infected files. Spoofing DNS servers is a common way for computer worms and viruses to spread [70,71].

#### 4.5.2. Attacks Based on Access-Level

There are two sorts of attacks, active and passive, depending on the attacker’s level of access to the system [58,72,73,74].

Active Attacks: In this case, the adversary has the intention to cause disturbance among legitimate nodes by impersonating or manipulating the routing information.Passive Attacks: In most cases, the intruder eavesdrops on the lawful transmitter’s and receiver’s communication in order to obtain the communicated data.

#### 4.5.3. Attacks Based on Transmitting Data

IoT devices employ sensors to detect and collect information about their surroundings. These sensors are susceptible to a number of concerns, including injecting fake sensor patterns or initiating commands to launch malware that has been placed in the device of a victim [75]. Attackers can leverage unintended communication channels between device peripherals to change important sensor settings (e.g., device mobility, temperature, pressure, light intensity, magnetic field, etc.) or send malicious instructions.

Transmitting via Light Sensors: One of the methods to compromise signals and send malevolent signals is light sensors which makes it simple to transmit data packets by turning on and off a light source. The difference of light intensity will act as a source of the device’s data flow. An adversary can activate malware by delivering a trigger message by regulating light intensity, i.e., adjusting the voltage of a light source.Transmitting via Magnetic Sensors: The functionality of a magnetic sensor depends on the magnetic fields of the surroundings of the device. It will be affected if there is a change in the magnetic fields of the device’s peripherals. To send data using magnetic sensor, permission from users is not required and can be silently run in the background [76]. An attacker can make forged data of the magnetic sensor by shifting the magnetic field of the atmosphere of the device. This action can ultimately trigger messages of malware. Triggering signals encoded by an electromagnet can be communicated to an IoT device, and this message will cause certain variations in the device’s magnetic sensed data. The triggering message can be derived from this electromagnetic signal by calculating differences.Transmitting via Audio Sensors: Malware in IoT devices can also be activated using audio sensors. Microphones in modern IoT devices may identify audio signals with a frequency relatively lower than the audible frequency range, and triggering messages can be sent using this type of audio signal to get over the device’s security measures.

#### 4.5.4. Attacks Based on Device Property

IoT devices are divided into two categories based on their device properties: low-end devices and high-end devices. Each kind of attack has a different level of effect on the devices. In these attacks, IoT devices might result in unusual behavior, or the device can be out of functionality [23,77].

High-end device class attacks: In high-end device class attacks, powerful devices with high processing capability launch attacks on the IoT network. Between comprehensive devices and the network, the internet protocol is utilized. The attacker uses the high processing power of the CPU of the powerful devices, e.g., laptop, computers, to connect with the IoT network to launch an attack from anywhere and anytime [78,79].Low-end device class attacks: It can be identified from the name of the attack that this attack involves low power and energy consumption devices to introduce an attack. The devices which are used in this attack can make a connection between the system and outside using radio links. A smartwatch, for instance, can be coupled to intelligent devices such as a smart TV, smartphone, smart refrigerator, smart temperature controller, and smart surveillance camera in a smart home system. It can also control the functionalities and configurations of these devices. Nevertheless, the smart home system can be under attack by low-power IoT devices like smartwatches [80,81].

#### 4.5.5. Attacks Based on Adversary Location

An intruder can launch an attack on an IoT system at any time and from any location. Someone from inside and outside can initiate an attack [23,77].

External attacks: Most external attacks happen with the aim of stealing the private data of users using malware such as worms, Trojan horse viruses, phishing, and the like. A hacker can gain control of the IoT device that is located anywhere on the IoT network. Without prior knowledge of the architecture of the network, the adversary tries to get access using a trial and error process. He will try until his attempt is successful.Internal attacks: An insider threat is someone who is a legitimate user of a network, system, or can access data and deliberately misuses it, or whose access leads to abuse. In this example, the attacker gains access to a component within the security IoT border and runs malicious malware against IoT devices. Compromised actors, unintentional actors, emotional attackers, and technology perception actors are the four categories of internal attacks.

#### 4.5.6. Attacks Based on Attacks Strategy

To initiate an attack, the intruder follows a strategy to insert and run his designed malevolent code in an IoT device to disrupt the IoT network. The attacker’s strategy can have two view aspects: physical or logical [77].

Physical attacks: To carry out this method, the attacker must have direct access to the IoT network’s infrastructure. The adversary can disturb a partial or full IoT network by changing the instruction or structure of the system.Logical attacks: Contrary to physical attacks, in logical attacks no physical access is required. Attackers do not cause any damage to the device physically; instead of this, the attacker makes the communication channel dysfunctional.

#### 4.5.7. Attacks Based on Information Damage Level

Sensors monitor and collect information from IoT devices. Attackers can change the information easily, which is fluctuating and open. Based on modifying the level of information by attackers, it can be divided into six categories [77].

Interruption: An interruption attack is an attack that creates an obstruction of any kind during the communication process between one or more systems. As a result, resource exhaustion is one of the outcomes of an interruption and the systems, which are under attack by illegitimate users, become unusable which results in the system being in shutdown mode.Eavesdropping: When a hacker uses unprotected or insecure data transmission to steal data shared or communicated through electronic devices, the attack is known as an eavesdropping attack. A sniffing or snooping attack is another name for it. In this attack, there are no anomalies in the network communications themselves, which makes the probability of this attack higher. IoT systems’ confidentiality breaches when there is an eavesdropping attack.Alteration: As the name suggests, in this attack the message which is sent by the sender is altered and transmitted to the destination by an unauthorized user. The integrity of the message and security requirements of the system is threatened by this type of attack. The receiver receives the exact message that an attacker is wanted to send. It will result in the poor performance of the network. To deceive the communication protocol, the adversary does it in an undisciplined manner.Fabrication: In this type of attack, an intimation data is inserted into the network by an unauthorized user as if it is a valid user who threatens the authentication of the IoT system. As a result, the message’s confidentiality, authenticity, and integrity are compromised.Message Replay: A replay attack (also known as a playback attack) is a type of network assault in which genuine data is captured, and the original message is resent or delayed in order to breach the target IoT devices. The IoT recipient device will be confused by a replayed message, which could harm the IoT system. This is one of the lower-tier variations of man-in-the-middle attacks.Man-in-the-middle: As the term implies, an attacker stands between the transmitter and the recipient; the attacker relay and possibly alter the communication between two parties, and the attacker collects information from the communication channel. Two parties feel they are conversing directly with each other in this situation.

#### 4.5.8. Host-Based Attacks

This form of attack targets the host’s resources, such as people, software, and hardware. OS and system software or applications are embedded into IoT devices. So, it is possible to attack IoT devices through the host of the IoT system. The adversary is presumed to have gained access to the victim in this exploit. Hardware-based attacks, software-based attacks, and user-based assaults are the three types of host-based attacks. The attacker compromises these resources and each categorized attack has a different impact on the IoT network.

Software-based attacks: Software can be compromised by an intruder to push the IoT device into a vulnerable state. An attacker can drive the device into an exhaustion state or resource buffer overflows by host-based attacks. For example, by compromising the sleeping mode, attacker can cause a sudden shutdown of the device due to low battery which breaches the interoperability of the system.Hardware-based attack: In this type of host-based malfunction, an intruder compromises hardware by cloning/tampering. Attackers steal actual driver, theft data by connecting to a device and injecting malicious code. For example, using an illegitimate charger which transfers malicious code along with charging, a smartphone can be exploited.User-based attacks: Sometimes, a user intentionally or unintentionally shares confidential data such as credentials (e.g., password, security, etc.) keys related to security. For instance, a user writes down a password in a file that is accessed by an unauthorized user who will be able to access the IoT devices [82].

#### 4.5.9. Protocol Based Attacks

In a protocol-based attack, the attacker can steal or change information by deviating or disrupting the deliberate protocol [58].

Deviation from protocol: The attacker impersonates a legitimate user and injects malicious code into the IoT system, causing it to deviate from the protocol. Application and networking protocols are the protocols where attacks happen to make a deviation from the protocol.Protocol disruption: The availability is one of the security characteristics in the context of IoT security. It is important to have functional security to make a reliable IoT system. However, attackers can disrupt IoT networks from both inside and outside the network, putting the network’s availability in jeopardy.

### 4.6. Classification of IoT Security Attacks on Different Layers of IoT Networks

Cybercriminals look for device vulnerabilities and exploit them to gain an advantage in conducting attacks, reinforcing the need for security from the start. Cyber attackers can challenge an IoT application in a variety of techniques that fall into four categories: physical or perception, network, software or application, and encryption attacks. Physical, network, and application-based attacks on an IoT system are all possible, as are attacks on encryption techniques. The classification of security threats on different layers of IoT networks is shown in Table 5.

#### 4.6.1. Physical Attacks

These attacks result from breaches to the IoT system’s equipment, such as the IoT device’s sensors, and attackers gain access through proximity, like inserting a USB drive. According to estimates, 70% of all cyberattacks start from the inside, whether deliberately or accidentally by humans. This attack can limit the lifetime or functionality of the hardware. We will discuss these attacks here.

Node Jamming in WSNs: A node jamming attack in WSNs is carried out by malicious nodes in the network, which interfere or disrupt or jam the radio signals used by sending useless information on the frequency band used [83]. An attacker can expel the service of the IoT device if the attacker can manage to jam the key sensor. In the case of a jamming attack, the statistical features (for instance, mean and variance) of a packet flow will fluctuate temporarily. Depending on its strength, a jamming source may be able to take down the entire system or only a small piece of it. This blocking can be temporary, periodic, or permanent.Physical Damage: A physical damage attack is an attack that results in physical damage to the targeted infrastructure. The attacker can gain access to the devices by physically damaging entities of the IoT network to serve his interest. This type of attack can be avoided if the protection of the zone where IoT devices are located is strong [84].Node Tampering: A node tampering attack is an attack where an adversary damages a sensor node. The entire sensor node can be replaced, or a part of the hardware can be compromised. A compromised node can be created by altering or replacing a node, and the attacker can control that node. By getting access, an attacker can alter sensitive information, for example, shared cryptographic keys/credentials (if any) or routing tables or other data, as well as disrupt the functionality of higher communication levels [20,39,79,85,86].Social Engineering: In social engineering attacks, malicious activities are accomplished through human interactions [87]. Social engineering is the psychological manipulation to trick users of an IoT system into performing certain actions or giving away confidential information that would serve their goals. Social engineering attacks may involve one or more steps. An attacker begins by gathering the background data needed to carry out the attack, such as potential avenues of entry and inadequate security procedures. The attacker then goes about gaining the victim’s trust, providing encouragement for additional behaviors that breach security principles, and obtaining the necessary information.Malicious Node Injection: In this attack [84], the attacker physically deploys a malicious node in the IoT network, which gathers information for the attacker or modifies the communication data to pass wrong information to the other nodes. Hence the attacker controls the transmission and reception of data flow and, finally the operation of the nodes.Sleep Deprivation Attack: The IoT devices are programmed to follow a sleep routine to remain in a low power mode for as long as would be possible without adversely affecting the node’s applications; hence it extends its battery life. The computing devices, such as a sensor node, which are powered by a battery, are susceptible to sleep deprivation attacks [88]. The attacker interacts with the node in such a way that appears to be permitted; however, the intention is to keep the victim node active which will lead to higher power consumption and become out of order [88].Malicious Code Injection/Forgery Attacks: The attacker attacks the device physically and injects malicious code to compromise the system and gain access. This can be done by inserting a USB with a malevolent program onto the node or by inserting communication links using methods such as (1) inserting malicious codes/data packets which seem legitimate; (2) modifying codes/data packets after capturing; (3) replacing the previously exchanged messages between nodes. The purpose of a malicious code injection attack could be various, e.g., to steal data, get control of the whole or partial system, and propagate worms [86,89,90].RF Interference on RFIDs: RFID is an auto-identification technology where communication is done using radio frequency (RF) with an identification code (ID). In this attack, the signal is compromised by creating and sending noise signals over the RF signal, which is being used for the communication of RFIDs. The noise will cause disturbance for RFID signals [91,92].Tag Cloning: In a clone attack, the attacker wants to have a tag that will have the same characteristics as the original tag and eventually can replace it [93]. In this attack, an attacker will be able to copy information of an RFID electronic tag or smart card to a cloned tag through reverse engineering or directly from its deployment environment. Sniffing, eavesdropping, and other technologies are used in clone attacks to get all of the data from the original tag, including encoding and customer information. It can copy all of the data to an RFID tag that can write to the entire region and replicate the tag [94,95].Eavesdropping: An eavesdropping attack, also known as a sniffing or snooping attack, is information thievery as it is transmitted using wireless communication. To eavesdrop, an attacker can use an antenna to get communicated data in an RFID system [96]. To be successful, an attacker finds a weak connection so that he can exploit it to reroute network traffic [97].Tag Tampering: In a tag tampering attack, the attackers’ aim is to alter the tag’s identity. The attackers can counterfeit a tag in an RFID system by using the method of tag manipulation. The attackers will get access to the communication channel by tampering the tag of an IoT device [63,98].Outage attack: An attacker sometimes can use more power than the allowed range or turning power of a group of objects which are placed in unattached environments. This operation will make the devices out of order [63,73].Object replication: As IoT devices are not monitored physically in remote places, an attacker, in this type of attack, can physically insert a new device/object into the network. For instance, a malevolent object could be added by replicating the object’s identity. As a result, such an attack could result in significant network performance degradation. Aside from degrading performance, the malicious object can simply corrupt or misdirect received packets, giving the attacker access to sensitive information and extracting secret keys [85].Hardware Trojan: A hardware trojan attack has been identified as the prime security issue in an integrated circuit in many different forms of research. Like other attacks, the aim of the attacker is to gain access and collect confidential information and firmware. In this attack, the adversary maliciously modifies the integrated circuit. The hardware trojan attacks are planned during the design phase and remain inactive until the designer sends it a trigger or an event [20].

Here we will show countermeasures against physical attacks. Table 6 shows physical attacks with compromised security goals and countermeasures [11,63]. Here, we are refereeing to compromised security requirements from Table 4’s abbreviation.

#### 4.6.2. Encryption Attack

Encryption attack is of making vulnerable to the encryption scheme which is being used in the IoT system.

Man in the Middle Attack: The man-in-the-middle attack is a cyberattack where the attacker takes a position between two users on the communication line and shares keys with both users. The adversary can intercept the signal that both users are sending to each other and can encrypt or decrypt data with the keys that he shares with both of them. An attacker can also alter the communications between two parties without their knowledge who think that they are sending data to each other [120].Side-Channel Attacks: Physical characteristics of IoT devices (e.g., power consumption, execution time, electromagnetic leaks, system fault, etc.) can reveal sensitive information. During the execution time of IoT devices, the intruder performs different tests to extract confidential information. Sometimes it is required to have technical knowledge of the inner working principle of the system, which will be exploited. In public-crypto systems such as RSA, extracting information from the device’s behavior is common [23]. RSA performs encryption and decryption of messages using private and public keys based on modular operations and large exponential values. The simple multiplication method is applied to modular operations, whereas the modular operations method is used on large exponential values. The attacker performs delay analysis to get how much time is required for calculating exponential results. Attackers can obtain private information like private keys by learning the computation time and using knowledge of the implementation technique. Most IoT devices will implement security measures like encryption to protect their sensitive information for security reasons. However, by performing a side-channel attack security mechanism can be broken.Cryptanalysis Attacks: The attacker in a cryptanalysis attack studies ciphertext, ciphers, and cryptosystems with the purpose of finding the encryption key being used by breaking the encryption scheme of the system. The attacker breaks cryptographic security systems and gets access to the encrypted messages, even without knowledge of the plaintext source, encryption key, or the algorithm used to encrypt it [121]. Secure hashing, digital signatures, and other cryptographic algorithms are also the targets of this attack. Based on the methodology used, there are different types of cryptanalysis attacks.-Ciphertext Only Attack: The attacker determines the plaintext accessing the ciphertext.-Known Plaintext Attack: The aim of this attack is to get ciphertext using plaintext. The attacker decrypts the ciphertext using the known parts of the ciphertext.-Chosen Plaintext Attack: In this attack, the attacker can choose plaintexts that are encrypted and find the encryption key.-Chosen Ciphertext Attack: Similar to a chosen-plaintext attack, an attacker gathers information by obtaining the decryptions of chosen ciphertexts. By utilizing the plaintext of chosen-ciphertext the attacker can find the hidden secret key used for decryption.

Now we discuss encryption attacks and corresponding countermeasures. Table 7 shows encryption attacks with compromised security goals and countermeasures [11,63]. Here, we are refereeing to compromised security requirements from Table 4’s abbreviation.

#### 4.6.3. Network Attacks

These attacks are focused on the network of IoT systems to extract large amounts of data remotely [97].

Traffic Analysis Attacks: Unlike eavesdropping attacks, the attacker does not need to compromise the original data, whereas the attacker listens to the network to gain some information using sniffing applications such as port scan, packet sniff, etc. Like eavesdropping attacks, in a traffic analysis attack, the adversary hears the communication flow of the network to analyze traffic to find critical nodes’ locations, the routing diagram, and even application patterns of behavior. Once an attacker reveals the required information, s/he can accurately host attacks like jamming, eavesdropping, sybil, etc., [126,127].RFID Cloning: Making a replica of a user’s RFID without knowledge is another way to overthrow RFID access systems. Even without accessing physically to the RFID card, an intruder can clone an RFID tag by copying data from the victim’s RFID tag onto another RFID tag [96]. The attacker can get the information and write the data to a similar blank RFID using off-the-shelf components by standing several feet away [128]. The integrity of the system will be violated as cloning results circulation of identical tags.RFID Spoofing: Unlike RFID cloning, in an RFID spoofing attack, an attacker does not physically replicate an RFID tag. Technically, cloning and spoofing attacks are made back-to-back. In this type of attack, an adversary impersonates a valid RFID tag to gain its privileges, reads, and records a data communication from an RFID tag. The attacker can obtain complete control of the system by posing as a legitimate source and sending his own data that includes the authentic tag ID. Spoofing attacks take place when a hacker successfully makes a position as an authorized user in the system [96,129].RFID Unauthorized Access: Different levels of security features can be available in RFID. If proper authentication mechanisms are not deployed in the RFID system, tags can be accessed by an attacker. The attacker can simply read, edit, or even destroy data on the RFID devices for his own gains. The attacker needs to execute complicated steps if strong level security measures such as access to the backend are required to retrieve the necessary credentials [130].Man In the Middle Attack: A perpetrator positions himself in an interface between two sensors, collecting private data and invading privacy by eavesdropping or impersonating one of the clients so that it looks like a normal information flow is taking place. The goal of this attempt is to obtain personal information that can be utilized for a variety of things, such as identity theft, unauthorized financial transfers, or unauthorized password changes. This attack relies exclusively on an IoT system’s network communication protocols. Therefore, physical presence is not necessary [94,131,132].Denial of Service: To carry out a successful denial of service attack, the attacker floods the IoT network with a large number of requests, resulting in a significant quantity of data traffic; this continues until the target cannot respond or simply crashes. In this attack, legitimate users are unable to use network resources to access information as all available resources are exhausted, which makes network resources unavailable to users. Moreover, many users’ unencrypted data can also be exposed [133,134].Sinkhole Attack: In a sinkhole attack, an adversary deceives the system by luring all data flow from neighboring WSN nodes into a metaphorical blackhole; the system is fooled into believing the data has already arrived its endpoint. The attacker uses a compromised node to attract network traffic by transmitting fraudulent routing information. The goal of the attacker is to breach the system’s integrity as well as disrupting network service. It prevents all packets from transferring, resulting in a sink or black hole in the network [135,136].Routing Information Attacks: In a routing information attack, the adversary uses a compromised node or a group of compromised nodes to make or change the routing information. The purpose of the attack is to obfuscate the system and make routing loops, permit or reject traffic, change the destination, provide fake error messages, shorten or extend source paths, or even partition the network; e.g., Hello Attack and Blackhole Attack [84,137].Sybil Attack: Sybil attacks are more common in networks with a large number of clients. A single node that unlawfully acquires the identities of numerous other nodes is referred to as a malicious node. The attacker uses the identities of the other nodes, causing the adjacent nodes to receive phony and incorrect information. The attacker can part in the distributed algorithm, such as the election where one sybil node has an identity more than once. It can also be selected as a part of the routing path, which can lead to a longer routing distance [138,139,140].Replay Attack: Attackers get information by eavesdropping on the messages of two parties, and the malicious node resends old packets to the overall system as broadcast or sent to a specific set of devices. When the other nodes receive these messages, they update their routing tables according to this expired information and reply regardless of whether the sender is transmitting any new packets or not. The Routing table and network topology will also be outdated, and with a huge number of packets replayed, both bandwidth and power will be consumed. This will result in the network’s activities being terminated sooner than expected, facilitating the impersonation attacks [20,86].HELLO flood attack: Some WSN routing protocols broadcast the “HELLO” packet to advertise themselves to their neighbors and construct a network topology. The attacker does not need to send legitimate traffic to conduct this attack. It can subsequently re-broadcast overhead packets with sufficient strength to deliver to every other network interface, leaving the network in disarray [141]. Though the malicious node is far away from network nodes, every node in the network will be convinced that the attacker is nearby. The majority of protocols that are impacted by this kind of attack depend on nearby nodes exchanging localized information to maintain topology or control flow.Blackmail attack: In the blackmail attack, a compromised node eliminates a legitimate node from the network by proclaiming that the legitimate node is a malicious node. If a compromised node is able to block a large number of nodes, the network will become unstable [142,143].Blackhole attack: In this attack, a malicious node, instead of forwarding all the packets, may drop those, and it may drop all the data traffic around the malicious node. This attack is also referred to as “Selfishness”. Its impact is highest if the malicious node is a sinkhole [110].Wormhole attack: A wormhole attack necessitates the collaboration of two or more adversaries with excellent communication resources (e.g., power, bandwidth) and the ability to construct better communication lines (called “tunnels”) between them. Malicious nodes are not clustered together; instead, they are carefully located at opposite ends of a network, where they can get messages and replay them in separate portions via a tunnel. Other nodes use the tunnel as their communication path and go under the scrutiny of the adversaries [144,145,146].Grayhole attack: A grayhole attack is an alternate form of a blackhole attack. The difference between blackhole and grayhole is dropping packet count. Instead of dropping all the packets like a blackhole, grayhole drops those packets it selects [110].

Here we will present countermeasures against network attacks. Table 8 shows network attacks with compromised security goals and countermeasures [11,63]. Here, we are refereeing to compromised security requirements from Table 4’s abbreviation.

#### 4.6.4. Application Attacks or Software Attacks

Software or application is the fourth factor that increases the danger of IoT security. Malware is introduced into the network’s application to start this attack. This malicious program can disrupt and monitor operations, spread viruses, corrupt or steal data, and more.

Virus and Worms: A virus is a small, self-contained computer software that can replicate and infect other computers in the same way that a real virus can. A virus is just one sort of “malicious logic” that can maltreat an IoT network. A computer worm is similar to a virus in that it spreads by itself rather than being disguised inside another software [170].Malicious Scripts: Usually, there is an internet connection in the IoT network. In this attack, an attacker fools the user, who controls the gateway, by visiting lucrative advertisements or websites and then running executable active-x scripts with malicious alterations to different parts of the system, which could lead to the system being shut down or data being stolen [171,172].Spyware and Adware: To propagate malware and harmful code, adversary attacks devices, such as IoT, use factory default user credentials, password brute-forcing, breaches, and manipulating weak configurations. Once the attacker has complete control, he can launch a DDoS assault against a target. A malicious party could insert harmful software into the system, which can result in data theft, data tampering, or even a denial of service [173].Trojan Horse: A Trojan Horse camouflage is a legitimate program where some previously defined event or date is present. This attack is triggered in the pre-set condition and then delivers a payload that may completely shut down the system [174].Denial of Service: An attacker can use distributed DDoS and DoS attacks to prevent users from accessing a system or network resource. This attack undermines the network or systems’ capacity to perform expected functions. This attack comes from numerous points, and it is difficult to defend against this attack [175].Distributed Reflection Denial of Service Attack (DRDoS): DRDoS is one form of DDoS attack. In this attack, an attacker preserves anonymity through an IP address by using a third party, called a reflector, which needs not be compromised [176]. Numerous victim machines that unintentionally take part in a DDoS attack on the attacker’s target are typically used in DRDoS attacks. Requests sent to the victim host computers are reflected back to the target from the victim hosts. The attacker sets the target’s IP address as the source IP address to the reflectors to flood the network with response packets. They typically also generate an increased volume of attack traffic [177]. The advantage of the attack is that the attacker’s identity will not be revealed as the attack will be done by reflectors, not by the attacker, which makes it hard to identify the original attacker and block the service. Another benefit of the DRDoS attack strategy is amplification. An attacker’s first request results in a response that is greater than what was sent when there are numerous victim servers involved. This increases the attack bandwidth and the likelihood that a denial of service outage will result from the attack. DNS, NTP, SNMP, and SSDP are the protocols for UDP-based DRDoS attacks; SYN and BGP are the protocols for TCP-based DRDoS attacks [178]. Firewalls, intrusion detection systems, machine learning, etc. can be used to mitigate DRDoS attacks [179].Firmware Hijacking: Hardware’s fundamental component is firmware. Every piece of hardware on a computer has this basic software preinstalled. Firmware modification or hijacking attack is one of the most catastrophic attacks where attackers can gain control of the entire system. It has already impacted various embedded systems such as telecommunication infrastructure, SCADA, and PLC systems [180].Botnet Attack: The botnet is a phrase derived from the concept of bot networks, where a bot is a computer program or robot that is automated. A botnet is a robot network. Usually, an attacker gains control by spreading a virus or other malicious code to take over a network of devices to create a botnet. This is not a malware attack, although it is carried out through compromising devices. An IoT Botnet is also a network of various malware-infected IoT devices, such as routers, wearables, and embedded technologies. This malware allows an attacker to control all the connected devices and eventually the network [181,182].Brute Force Password Attack: Brute force password attack or BFA is a search and find a method to gain privileged access where the attacker guesses possible combinations of a targeted password until the correct password is discovered [183]. Based on the length and complexity of the password, both time and the applied combination will be required. BFA is a password research technique that uses a variety of probable ASCII characters, either alone or in combination.Phishing Attacks: In this attack, the attacker gets private data like usernames, and passwords by email spoofing and phishing websites [184,185].

Here we will show countermeasures against application or software attacks. Table 9 shows application attacks with compromised security goals and countermeasures [11,63]. Here, we are refereeing to compromised security requirements from Table 4’s abbreviation.

## 5. Security Mechanisms

In general, the two approaches for protecting IoT and IoE devices from potential intrusions are software-based and hardware-based. Software is responsible for protecting devices against software-based attacks. It is tough to break the mathematical algorithm of software using the present computer system. However, it will be able to solve mathematical keys within a shorter time compared to the current approach when quantum computers become reality [199]. In software-based security solutions, devices are prone to attack as keys are saved in the non-volatile memory (NVM) of the devices. The invention of quantum computers might make software-based security solutions vulnerable. Therefore, the hardware-based solution could be one of the possible solutions due to the risk factor of existing software-based security [23]. This is accompanied by the prediction that the majority of current asymmetric cryptography will be broken by the advent of quantum technology employing Shor’s Algorithm [200]. Winternitz One Time Signature (WOTS) Scheme [201], Supersingular Isogeny Diffie–Hellman Key Exchange (SIDH) [200], etc., are post-quantum resistance. It is required to standardize a model to preserve integrity. NIST is working to standardize cryptographic mechanisms to resist attacks on the post-quantum area. Figure 10 shows hardware-based security mechanisms.

Cryptographic functions can be used in hardware-based security by using a dedicated hardware-integrated circuit or chip to store the keys. Physical computing devices called hardware security modules can be utilized for crypto processing and robust verification. They can encrypt, decrypt, save, and handle digital keys using various encryption algorithms, as shown in Figure 10 [202]. (1) a public key and (2) a private key; these two keys are carried by each node in the network. Man-in-the-middle attacks are easy to launch using this hardware-based security mechanism. Attackers can clone the device after stolen. PUF is a solution to avoid the attack [203,204]. Gassend et al. first introduced PUF as a security primitive for hardware-based security [205]. PUF can produce digital fingerprints on the fly when requested. PUF produces a different key (Response) when different input (Challenge) is fed to the device, which is due to manufacturing variation of the chip [206,207]. Now, the characteristics of PUF will be discussed. Figure 11 depicts the characteristics of PUF [208,209].

Uniqueness: It is required to have different responses for each challenge. It is determined by how distinct one chip is from another in terms of strings and ultimately process variation of PUFs. Hamming distance (HD) calculated with each response represents uniqueness. The number of bit differences between the two responses calculated the HD. Ideally, uniqueness should be 50%.Randomness: It is calculated based on the total count of “1”s and “0”s of a response. In response, it is expected that the presence of an equal number of “1”s and “0”s. If so, the PUF will have 100% randomness.Correctness: A PUF needs to generate the same CRP irrespective of external variables such as temperature, pressure, time, etc. In every operating condition, the ideal value is 100%.Reliability: A PUF, as the name implies, should be 100% reliable. Every time a challenge is presented, an ideal PUF will produce the same response.Uniformity: This measures how random the PUF is. The probability of “1”s and “0”s should be equal so that any intruders are unable to guess.Bit Aliasing: How a particular response bit across several chips are measured by bit aliasing, and the ideal value is 50%.Steadiness: All the responses should be identical when the same challenges are feeding to a PUF. Ideally, the value of intra-HD should be 0%.

PUF includes ring oscillator PUFs, Arbiter PUFs, Optical PUFs, SRAM PUFs, DRAM PUFs, and MRAM PUFs, among others.

Blockchain is a distributed, immutable, decentralized, and shared digital ledger that can be used in IoT and IoE networks to provide security [210,211]. There is no centralized entity to govern the data, and it is a peer-to-peer connection, unlike other systems. All nodes in the network come to a consensus to verify transactions and store the data in a block with a timestamp after the completion of the mining process by the miner node. The blocks create a chain which is called blockchain, and it uses SHA-256 and ECC for data integrity and authentication. Figure 12 shows the basic structure of a block in blockchain [212,213]. Each block stores timestamp, transactions, and previous block hash along with others’ attributes. It is very tough to alter data in blocks as each block is connected by a hash.

## 6. IoT Security Solutions

This section will evaluate security measures for the IoT network based on published literature. The best understanding is used to present each design.

### 6.1. PKI-Based

Guo et al. in [214] proposed a scheme to provide security for big data collection in large-scale IoV. Mutual authentication and single sign-on-based authentication schemes were developed for collecting big data in a secure way. It assumes that each vehicle is loaded with a certificate by a certificate authority before registration. During registration, if the certificate is valid to the data center, it will be registered as a valid account, and a private-public key will be issued. It uses sign-on to start data flow. If the message is from a valid account, then a session key will be generated. The private-public key with a signature is used for encryption. If it passes the RSU area, then it needs to update the session key with the new RSU. If RSU finds the signature of the data center and ID in the certificate and also receives within the expired timestamp, then RSU will update the session key. It does not hide the ID of the vehicle. After changing the RSU area, a forgery vehicle can login using a fake certificate.

In [215], Li et al. proposed a certificateless conditional authentication protocol to avoid the storage of certificates in TA and vehicles. Each vehicle has two unique tags which are linked with its secret key. Instead of using the secret key directly, two random values are used with the secret key to make the message unlinkability. It uses the discrete logarithm (DL) problem [216], and computational Diffie–Hellman (CDH) problem cryptography algorithm [217] and it assumes that the DL problem is intractable. In future work, they will work to remove vulnerabilities and limitations of this work to achieve security and privacy requirements and incentivize the participants.

Kerrache et al. proposed a solution based on the social network in [218]. It will calculate trust values based on online social networks for drivers and passengers. It also takes into account inter-devices authentication for in-vehicle, inter-vehicle trust calculation based on trust ratio, and also RSU trust calculation for each vehicle by getting recommendations from neighbor vehicles regarding a vehicle. Furthermore, it takes into account followed path by users and current mobility. It uses Chaotic map-based Chebyshev polynomials for computing security keys for inter-devices authentication, and social network trust is identified using the Advogato trust metric. The human honesty factor will be required if nodes’ behavior is unclear/compromised. This scheme required trusted third parties such as social network platforms, network providers, etc. It introduced delay to compute human factors and location-related trust.

Al et al.’s work [219] shows secure performed enriched channel allocation, i.e., shared channel, using commutative RSA (CRSA) [220]. In the CRSA scheme, two prime numbers are required to be sent to all vehicles using random number encryption and decryption parameters, which will be generated, and it is required to exchange keys to each vehicle. Data will be sent to the destination vehicle after encrypting in several vehicles by their keys, and the destination vehicle will decrypt the data using each vehicle’s keys.

Meshram et al. [221] also proposed a secure smart city communication protocol using a smart card, password, and extended chaotic maps. The protocol used random number with Hash functions (31 times) to make it lightweight. It contains processes to change password and revocation of smart card. If the smart card is stolen and the attacker can guess the online password through the side-channel attack, the smart city could be compromised.

### 6.2. ABE-Based

In [222], Han et al. proposed a ciphertext-policy attribute-based encryption (CP-ABE) system to protect communication. It uses a max-miner association rules algorithm to mine the frequency features to build frequent item sets. The ECU can access and decrypt the data if it has the same set of frequency attributes. To improve speed, it uses symmetric encryption in the registration stage. A secret key that is used for decryption is generated using the keygen algorithm, which takes frequency attributes as input. It only works with the same attribute sets. There could be a chance of a major attack if one ECU is compromised where an adversary can get the algorithm for a specific attribute set.

Hwang et al. [223] proposed a CP-ABE-based authentication framework for the healthcare sector. The protocol identifies which device received the first key (by the collaboration of attribute authority and trust authority) to identify the root cause of vulnerability. The length of the ciphertext is independent of the number of attributes, resulting in the same decryption time irrespective of the number of attributes. However, the proposed approach necessitates a significant amount of computation in order to validate the user’s identification. PHI leaking from the client who got the delegated key is likewise a problem.

### 6.3. ECG-Based

Huang et al. [224] proposed an ECG-based [225] authentication protocol for the IoMT application that uses Singular Value Decomposition (SVD) to de-noise signals of ECG. Interference will be decreased depending on the movement situation and pre-defined attribute frameworks. In the instance of gentle exercise, a de-noise signal was produced using weighted online SVD. Running and walking requires a lot of angular distance; additionally, different activities require different routines or stances. In this study, it was assumed that the adversary had no insight into the patient’s ECG template.

### 6.4. MAC-Based

Siddiqi et al. proposed a MAC-based [226] authentication system in [227] for the medical sector. The designed framework was a smart card and IoMT device authentication system based on public-key cryptography. The absent k-bit in the server’s shared hash function must be calculated and identified by the user device. However, the suggested approach lacks user anonymity.

In [228], Hahn et al. highlighted the security flaws of MAC-based and commitment-based authentication protocols (where a value will be used rather than a range). They presented a technique to improve the system in which a key server generates commitment keys and verification keys. After calculating a commitment value with those keys, the client will transmit the message. To validate the commitment key, the specialist will use the commitment key and the verification key to decrypt the partially decrypted content.

### 6.5. ECC-Based

Fog server-based authentication scheme is developed by Wazid et al. in [229]. Secret information is stored in the memory of RSUs, Vehicles, and fog servers from TA. It uses ECC-based key management. It uses XOR and hash operations for developing the authentication. To replace compromised nodes, it has a phase of dynamic node addition.

Wu et al. proposed an ECC encryption-based batch verification algorithm that has the main objective of reducing verification time [230]. It has been found that in this batch verification scheme, ten times less time is required compared to RSU only verification. When traffic is higher, then RSU allocates assistance verification terminals (AVT) based on computational power and network traffic. After verification, AVT shares a confirmation message which is finally verified by RSUs.

An ECC-based certificateless aggregate signature authentication mechanism was presented by Thumbur et al. in [231]. Combining several unique signatures on distinct messages from different vehicles into a single signature, it decreases verification time and storage requirement at RSU. It assumes the elliptic curve discrete logarithm problem is not forgeable. The vehicle generates public-private key pair upon receiving a partial secret key.

Zhang et al. also developed an ECC-based protection protocol in [232]. It calculates its private key by summing a stored key in the TPM module and a random number to avoid a side-channel attack. RSU broadcasts the certificate every five seconds, and the vehicle verifies that and shares the signature. RSU verifies the signature for mutual authentication. TA uses two hash functions to revoke the authenticity of RSUs and vehicles. However, as each vehicle generates its own pseudo-identity, the system will not authenticate the vehicle if the PID is in the compromised list. Also, this scheme needs secured communication between RSU and TA. Here, TA works as a cloud as well.

Ghahramani et al. [233] proposed an ECC-based framework to enhance the system proposed by Li et al. in [234] for mobility networks. Mobile users (MU) are connected to a home agent (HA), and when MU goes beyond the coverage area of HA, it will be connected to a foreign agent (FA). The storage cost of the framework is high as MU, HA, and FA need to store many parameters. Moreover, each FA needs to register with each HAs and also needs to have information about MUs. FAs will complete the authentication with HAs when MU makes a request. Compared to other frameworks, this requires many more calculations.

In [235], Xie et al. solved the flaws of the framework of Banerjee et al. [236]. This work covers the communication between the sensor and the user via a gateway, but the communication to the server is absent here. Verification of user in sensor should be done in step two instead of step four to avoid unnecessary computation and communication in case of invalid user. The protocol could suffer from a smart card stolen attack.

Xia et al.’s [237] proposed a framework to work on edge data of smart grids. To secure against reply attacks, random nonces were encrypted in each stage. The smart meter used the private key to prove its identity. It is required to consider security at the hardware level.

Another authentication framework was designed by Chen et al. in [238]. The method covers the communication between edge devices and the smart meters. The secure communication to the control center is not presented. It is unclear how the smart meter will select edge nodes based on hashed pseudo-ID. Edge nodes could be compromised if an attacker could extract the pseudo-IDs of the edge nodes. The authentication is completed in two message flow. Srinivas et al. in [239] showed an authentication mechanism using ECC for the vehicular network. The method needs to be optimized as it requires high computational cost and communication overhead due to complex calculations.

Velliangiri et al. [240] proposed an ECC-based authentication method for Industry 4.0. The method has four steps which are initialization phase, certificate registration, certificate publication phase, and data encryption phase. The article did not provide proper utilization of every parameter used in the initialization phase, and the paper needs to show elaborate processes. Moreover, the computational cost is around 8 s, which needs to be optimized.

Ryu et al. [241] also proposed an ECC-based authentication framework for the medical system. The scheme used ID, password, biometrics, and a random number as features to generate different parameters to store for the authentication phase. If an attacker gets stored parameters, s/he will not be able to complete authentication due to the absence of features. But the framework can optimize storage cost, and computation cost as two parameters do not take part in the authentication. Moreover, the computation and communication cost is high compared to other frameworks.

For smart parking, Khaliq et al. [201] proposed an authentication method using ECC and location differential privacy based on Laplace distribution. The method needs to use third-party app and also hide the sensitive data in local settings before sharing with service provider. The method needs to change pseudo-identity after each verification which is not presented in the paper.

In [242], Ali et al. proposed a method for V2I communication. In the method, vehicle used identity based cryptography to signcrypt plaintext, and RSU used PKI to de-signcrypt the messages to get the plaintext.

### 6.6. ML-Based

It is a supervised machine learning-based data-centric misbehavior detection model development that was proposed by Sharma et al. in [243]. Scikit-learn was performed to complete the experiment. This work includes six algorithms to identify position-based attacks. Using supervised ML, four quantitative features are extracted to check location plausibility and movement plausibility to find out patterns and predict the misbehavior of vehicles.

Pascale et al. used in [244] an embedded intrusion detection system in an SoC in the vehicle. This work checks whether present data or received data is malicious by initially filtering all the messages on the controller area network (CAN-Bus). Then it passes two steps algorithm to detect attacks. In the pre-processing step, it analyzes ten state frames containing several parameters, such as RPM, brake, speedometer, etc., recorded with timestamps in each 4 ms through spatial and temporal analysis. The Bayesian network, uses Bayes’ theorem where a probabilistic graph that predicts the dependency relationship using a set of random variables through a probabilistic inference process, is applied in the second step. It identifies attacks by comparing previously trained through a pre-established data set and parameters obtained as information from these parameters.

### 6.7. CHAP PPP-Based

Challenge-Handshake Authentication Protocol Point-to-Point (CHAP PPP) based authentication protocol proposed by Pradeep et al. in [245] for the smart city. CHAP guards against replay attacks by changing the challenge. The secret parameters must be in plain text to both client and server to complete authentication. It is a three-way handshake where challenges are sent using hash and the other side compares with the predicted value. One challenge is that the same challenge cannot be used at the same time by other clients. If the password or the secret key is revealed, the client will be compromised. It is highly unacceptable to physical or side-channel attack.

### 6.8. PUF-Based

Wang et al. in [246] designed a novel on board unit (OBU) with three-level security layers. The architecture of OBU is divided into four areas which are (1) Area A: Core security area which is assumed to be secured, (2) Area B: Signal transmitting area which takes input from other areas, (3) Area C: External hardware devices access area which communicates with authenticated devices, and (4) Area D: External network access area where network interface cards and PUF are embedded. The authentication scheme works in three stages which are (1) Periodically updated session keys to authenticate external networks and devices, (2) PUF for authenticating multiprocessor and information from internal area B, and (3) Hardware isolation using FPGA logic gates.

Alladi et al. developed a scheme using both ECC and PUF [247]. TA is responsible for registering each vehicle in the network. In the registration phase, a secret key is generated using ECC output and the Vehicle ID. Unlike other PUF-based technology, it uses one CRP for each vehicle. Each vehicle is registered to the cloud server. For authentication, the vehicle raises a request to RSU, and RSU validates vehicle ID using CRP with the help of the edge server and also checks timestamp difference. The secret, along with nonce and other parameters, are exchanged between the vehicle and RSU for completing authentication. It is required to have a secured channel between RSU and the edge server.

Aman et al.’s work considered an RSU gateway to serve several RSUs, and once the vehicle is authenticated with an RSU of an RSU gateway, the vehicle does not need to be authenticated with other RSUs of that RSU gateway [248]. A vehicle sends its crypto-identity, constructed using a secure hash of ID, secret PUF response, and a random nonce, to RSU for authentication purposes. Every RSU has a stored challenge like Alladi et al. to generate a response, and the RSU encrypts received messages from vehicles using generated response and sends it to the RSU gateway. RSU gateway verifies the response and extracts messages and shares with TA using a secret key. TA generates a token that contains the session key along with CRP so that vehicle can extract the session key using CRP. This session key is being used to communicate with other RSUs of the same RSU gateway. It is required to update the CRP of each vehicle with TA periodically.

Alladi et al. also proposed a PUF-based authentication scheme in [249]. During registration, it stores a CRP in the server and the challenge to the device memory. For authentication, the device generates a response to the stored challenge and the server compares it with the stored response. Server verification is done by diving the response and computing with a nonce. If the device gets the same response it will then generate a new CRP and will share that with the server and the server will replace the previously stored CRP. A new temporary ID and session key will be generated. It could reveal the vehicle ID as it sends the ID in a message. There could be a server impersonation attack if an adversary asks for a response from the vehicle.

In [250], Yanambaka et al. presented a PUF based security solution for IoMT. A PUF based on a hybrid ring oscillator was created for the authentication mechanism in this study. CRP was saved on the server, and the IoMT device was authenticated depending on the device’s answer to the server’s challenge. It can withstand client impersonation attacks, but it cannot withstand machine learning or server impersonation attacks.

For secure communication in a smart grid, Badar et al. [251] proposed an authentication scheme using PUF. PUF is generally used to avoid storage in device memory, but the framework needs to store in edge nodes. CRPs are used for control center verification in the last stage. The protocol is not free from physical attack, and the communication cost is high.

Tanveer et al. [252] used ECC and PUF to make reliable smart grid communication. The PUF was used to decrypt the secret keys in the smart meter. As the secret key is stored using encryption, the attacker can get the data using physical access and tries to decrypt it. If the attacker is able to decrypt, it will be able to impersonate the smart meter. Moreover, the framework has some unnecessary storage and computation. If the PUF is reliable, it will be able to produce an exact decryption key to get the secret key, so it is not relevant to verify the combination of key and response.

Lee et al. in [253], proposed a PUF-based dynamic group authentication key agreement framework for IoMT. The framework has medical device addition, rejection, fault identification, and tolerance functionality. Though it uses PUF to avoid storage, it needs to store a parameter that will be required for computation in the authentication and group key agreement phase. Authentication and key agreement between two devices are started through the register center by sending a seed. After that, two devices authenticated each other by generating PUF response, stored parameter, hash, and XOR operations. After mutual authentication, both devices broadcast a secret key, which helps other to update the group key. If the mutual authentication between two newly added devices before communicating with existing devices, then the group will not be able to update the group key. Also, how the new device will get the existing group key needs to be clarified.

### 6.9. Blockchain-Based

Jiang et al. showed the application and performance of blockchain in the case of IoV in [254]. It presents how data flow and re-transmission are related to network traffic growth. Security details will be considered in their future work.

To make faster transaction confirmation and speed up new users, this work established a novel debit-credit mechanism for blockchain-based (PoW) data trading by Liu et al in [255]. A vehicle can loan from multi vehicles as per demand by promising to pay interest and reward. It uses a two-stage Stackelberg game to maximize the profits of borrower vehicles and lender vehicles jointly. Nash equilibrium is reached when there are optimal pricing strategies. To make the account secure, it uses encrypted signatures using corresponding keys and certificates. It is assumed that an adversary cannot forge a signature of a vehicle or gain control over the majority of system resources.

A blockchain-based incentive mechanism has been proposed by Yin et al. in [256], where multiple vehicles can participate in bidding to allow their resources to complete a task through a smart contract for secure transactions. The method identifies malicious bidding by unusual price and quality of data provided. If any vehicle is identified as a malicious vehicle, then it will be removed from the network and it will not get any reward. For accomplishing any emergency task, the bidding process will be omitted, and multiple vehicles can make a cluster as per the same time allocation and resources. A novel time-window-based method is developed to manage the emergency task. Here, as malicious nodes are identified after receiving a report, a malicious incident can happen. Also, in case of an emergency task, it ignores the bidding, and the communication of inter-vehicle for clustering is not depicted which can compromise clustered vehicles and networks as well.

Yang et al. proposed a decentralized trust management system based on blockchain [257]. It employed a joint PoW and PoS consensus mechanism. Vehicles validate received messages from surrounding cars and calculate credibility using the Bayesian Inference Model. Vehicles consider the distance between sender and event location. Based on the result, vehicles upload ratings to neighboring RSUs and it then calculates trust offset to finalize data. RSUs work as a miner. It assumes that RSUs will not be compromised due to the limited resources of an attacker.

Gao et al. combined 5G network for low latency, a software-defined network (SDN) for effective network management, fog computing for avoiding frequent handovers, and an incorporated trust model to make the decision about message authenticity in [258]. The vehicles are equipped with SDN-enabled OBU. The practical Byzantine fault-tolerance consensus algorithm is used for verifying correctness. RSU hubs are used as miners, and PoW puzzle solutions are used in this blockchain method. In this method, peers provide verdicts about the trustworthiness of the information, such as time of event reporting, location of the event, etc. provided by the neighbor vehicle. The trust model is organized into three algorithms, (1) Cluster model: neighbor vehicles make a cluster, (2) Message forward model: Verdicts are collected and aggregated and forwarded the message to the other clusters in the network to create awareness. (3) Judgment model, which concerns the decision obtained over a sender’s message after several verdicts. However, a Cluster should be random as vehicles are moving and no pattern follows.

Xu et al. proposed an energy-efficient transaction model using blockchain in [259]. This work mainly focused on fewer transactions to reduce network traffic. It used asymmetric encryption to incorporate security. Instead of sending direct data, it will send functions using an adaptive linear prediction algorithm where a certain amount of real data will be trained. If the new data is not within the preset value, it will forward the function. By sensing energy consumption/rewarded coin system, the system will sense whether devices are compromised or not. Data accuracy depends on the amount of data. If there is a high amount of data, the error rate will be lower.

A blockchain-based scheme was proposed using smart contracts for securing the framework for registering trusted vehicles and blocking malicious ones. The scheme used certificates for communicating and reserving privacy with vehicles and RSUs, PUF for establishing trust, dynamic PoW consensus algorithm for scaling incoming traffic by Javaid et al. in [260]. RSUs act as miners. In the registration phase, CRPs are stored, and a certificate is issued. If the vehicle is in the local blockchain, then the certificate will work for authentication. In the authentication phase, vehicles and servers share CRP for trust establishment. After that MAC is used with the stored parameter in memory for ensuring data integrity. After verifying MAC, it uses the private key to the hash, and then a certificate will be used instead of the CRP of PUF later. It is considered that RSU will not be compromised. As RSUs store certificate and it is prone to attack, the system could be compromised. It can reveal few responses to an adversary as it is sharing a response after getting a challenge. It is also using private and public keys hashing.

A blockchain and ECC-based authentication framework for the smart city was proposed by Vivekanandan et al. in [261]. Mutual authentication between two devices happens by sharing the stored secret key in the devices. The method excludes the usage of gateway nodes to reduce computational cost and also uses private blockchain for registration purposes which can only be accessed by authorized personnel. It used the location as a feature for authentication purposes. The method did not use central authority during authentication except at registration time, but the method does not require any information from the central authority. So, the registration information of the central authority is irrelevant. Moreover, it says that ID is a permanent secret, but it uses the ID of other devices during authentication. An attacker can eavesdrop on the message flow and can get the secrets to impersonate in the future. Furthermore, the authors did not describe the authentication framework which is required for the readers to get a full view.

In [262], Wang et al. pointed to the rise of heterogeneous data (for example, for ehealthcare systems) that is being used to transfer large amounts of patient records to centralized cloud servers for illness diagnosis. It is, however, vulnerable to a number of security problems that can be mitigated by authentication. In the paper, the authors suggested an authentication scheme that uses blockchain technology and PUF. In addition, a fuzzy extractor approach is used to deal with biometric data. Their investigation revealed that their work has the smallest computing and communication costs of all the systems considered.

Son et al. [263] proposed blockchain-based authentication framework considering handover for V2I communication. In the method, ECC is used to perform the initial authentication. To avoid complex computation at the time of handover, the method used only hash and XOR operations. RSU is responsible for authentication, and it used signature to validate the transaction. Some clarity is required regarding few parameters as those are not showed as stored or calculated. Moreover, the method could be affected by smart contact capture and dictionary attack if a verification can be avoided in the vehicle to initiate authentication.

Yang et al. in [264] proposed multidomain authentication using blockchain to build distributed trust. There are three layers: the first layer is the perception layer where vehicles, RSU, etc. will be present, the management layer is the second layer where certificate authority and third-party authorities will form a consortium; and, the third layer is the blockchain layer which stores credentials, records, cross-domain information, etc. It introduced a key derivation algorithm to generate batch pseudonym distribution to resilient key escrow. In the future, the authors will incorporate V2V secure confidential data sharing.

Cheng et al. [265] incorporated blockchain, certificateless cryptography [266], ECC, and pseudonym-based cryptography (PBC) [267] to have a secure authentication between edge servers and IoT devices. As identity-based encryption (IBE) is vulnerable to key abuse issues, certificateless cryptography is introduced, which will generate a public key using the identity of the device and a secret value only known to the device. PBC is based on IBE to hide the identity. The method has static mutual authentication for static devices, intraedge mutual authentication for moving devices from one server to another server, and interedge mutual authentication for mobile IoT devices. After authentication, both the IoT device and the edge server will decide the new session key for encrypting shared data.

Xu et al. in [268] proposed a blockchain-based group key agreement protocol where a device needs to authenticate a device that is in the left position of it. A key distributor center (KDC) will store the credentials and parameters of the serving group. After collecting parameters from the blockchain, the right side device will authenticate the left neighbor. After authentication, using group authentication, the group key will be negotiated. The framework could be affected by device capture attacks, and a malicious user can add more devices as only the right side device can authenticate the new device, and the authentication will only be performed for a single time. Also, the process of changing the location of serving KDC is not mentioned.

In [269], Xu et al. proposed an authentication framework using blockchain and token mechanism. After authentication, a device will get a token that will be valid for a certain timezone. A device will get a token from a private key generator (PKG) which forms the blockchain. All the PKG will form a mesh to support in case of any failure. After the completion of authentication, the group will decide the group key. In the future, the authors will include more security features and will reduce energy consumption with less complexity.

Table 10 shows a comparative analysis of IoT security solutions.

## 7. Challenges and Future Directions

Although there have been some difficulties incorporating IoT and Wireless sensors into Industry 4.0, smart systems can address a variety of issues encountered by the industrial sector. IoT and Wireless sensor technological advancements have raised more questions regarding security, privacy, and data management. Manufacturers and businesses struggle to effectively manage the growing amount of data being produced. AI algorithms are being used to handle Big Data and improve the intelligence of systems and gadgets. The data is processed using the algorithms throughout a variety of time frames.

### 7.1. Lightweight Cryptographic Algorithms

Data leakage can be greatly increased by the billions of connections that exist between equipment, people, and information-sharing network. IoT systems are so prevalent and pervasive that there will inevitably be issues with network privacy and data security. Data safeguard, and information security must be ensured in a variety of activities, such as transportation services, personal affairs, operations, and information protection. Encryption is one of the approaches to do so. Failure to use encryption could result in data being intercepted, changed, or even permanently deleted. As a result, encryption methods—more specifically, dynamic encryption need to be used to protect the data and guarantee its confidentiality. Different attacks, such as insider attacks can not be resisted using encryption, and it can cause massive issues to the IoT network. Symmetric key based cryptographic methods might not be suitable for low memory devices. Similar level of security can be provided by ECC with lower size of the key. To avoid storage, PUF can be incorporated as a cryptographic method. PUF can produce unique keys. However, PUF can be impacted by external factors, such as voltage, pressure, humidity, noise, and many other factors. Reliability is a major issue for PUF for being deployed. Moreover, developing a strong PUF by maintaining all the required characteristics is difficult. Furthermore, different research showed that PUFs are susceptible to modeling attacks. XOR PUF shows better resistance compared to arbiter PUF, and a few others. ML is also used as a security primitive. ML is mainly used for intrusion detection rather than as a cryptographic algorithm. The cryptographic measures will raise transmission overhead that can be considered as a burden for IoT applications such as IoMT. Latency reduction and resource utilization are major concerns for IoT applications as real-time data communication is required for monitoring and decision-making in different IoT applications. Moreover, processing huge amount of data requires battery power. If the power consumed cryptographic algorithms are applied that could end the battery life and impact on life-cycle of the devices.

### 7.2. Lightweight Authentication Frameworks

One of the major elements of IoT is security and privacy. It is becoming a major challenge to ensure the privacy of user data. If an attacker can enter the system, s/he can find out the way of doing DoS or MITM attacks by getting a clear idea of the communication framework. The level of vulnerabilities will be increased further if the system is in an open environment rather than a close environment. This article gave a brief overview of existing authentication frameworks for different IoT applications. RSA based authentication scheme needs more key size length compared to ECC based mechanism. Feasibility checking of ABE and ECG based methods need to be carefully designed for the IoMT network. Attributes will be changed from time to time and it will affect the authentication. Like ABE, ECG parameters may fluctuate based on the health condition. By physically accessing the device, an attacker can get the encryption key which can be used to get data of future communications. On the other hand, direct uses of the PUF key can be subjected to modeling attack. Many PUF based authentication methods require storing challenge which is contradictory to PUF benefits (removal of storage). Moreover, PUF based methods also require a separate module or logic gates to be embedded in the device which may raise complexity or increment the size of the device. There might will some impact on prices as well. Furthermore, to avoid server impersonation attack, researchers used PUF in cloud server as well. As server can receive multiple requests at a time, how the server will process the request using PUF is not clarified. Also, PUF increases the database size of the cloud server and for a large network, server needs to have additional resources to make query to the database. Different frameworks use many authentication steps and complex calculations to complete the authentication process, which increases computation cost and communication overhead. Authentication protocol needs to be designed with fewer numbers and sizes of messages with simple calculations. Instead of using static structure, dynamic structure can provide same level of security with less data exchange. FL can be a good option for authentication, which uses a local model to the update global model by cooperative training and getting data. Also, the power of AI can bring an advantage to complete the authentication process. Most of the researchers focused on vehicle to server secure communication. It is required to put focus on the vehicle to others’ communication to make a secure and complete IoV environment. Mobility is also required to be considered for IoV or other applications as vehicles/devices might move outside of the current serving region. While considering authentication framework, cloud security needs to be ensured for centralized methods, which is a major research area. Cloud could be public or private. Cloud storage is vulnerable to data manipulation and illegal resources, which could interfere with the process of smart networks. Decentralized authentication can be applied for both homogeneous and heterogeneous networks. Elements of blockchain network such as smart contract need to be robust enough to maintain security. With the introduction of 6G, more devices with large data will be involved in the network. To ensure the authenticity and reliability of information, data anonymity needs to be addressed in the form of cipher, hash, random nonces, masking, PUF, etc.

### 7.3. Data Processing

A large amount of connected IoT devices generates enormous data called Big Data. The sensors and actuators on millions of consumer devices are interconnected, interacting, gathering, and transmitting the information. Each device produces or senses a large amount of data that requires further processing. The main problem will be making sure that these different forms of data are handled appropriately and efficiently, especially given the lack of time, computational resources, and processing power. In 2000, the volume of stored data was 80,000 petabytes, whereas it was predicted that by 2020 the volume would be 35 zettabytes. However, only 20% of these are analyzed by traditional methods due to different data structures, and the rest 80% is not taking part in the decision-making purposes. It is assumed that the quantity of data will be increased further with the evaluation of 6G. Data-driven methods can be developed with the help of FL and AI to handle Big Data from large-scale IoT devices.

### 7.4. Scalability

There are a remarkable amount of devices that make up the IoT. Instead of being networked as a loop, such devices are typically connected to each other in hierarchy subdomains. As a result, there are substantially more connected devices than there are on the present internet. Scalability is hampered by the architecture’s complexities. In order to address complex networks and maintain extremely simple techniques in the network, this problem should be solved. A system that supports and scales with a wide range of sensors recognizes and addresses this. Moreover, authentication frameworks should be designed in such a way that they can be deployed in a system with a large number of resource-constrained devices.

### 7.5. Interoperability

A variety of sensors featuring various sensing methods or communication spans are found in a heterogeneous network. Researchers should undertake the required analysis compared to isomorphic situations because it is feasible that heterogeneity influences the problem formulation. In a network, there will be devices from multiple vendors, which need to ensure communication among those devices. Moreover, a system needs to establish connectivity with another network of different functions which can not be ensured without interoperability. Security and privacy are the major concerns while data traverse from one network to another. The data structure should be uniform to process the data and make decision.

### 7.6. Governance

IoT is being accepted in every arena, which demands governance to set goals of organizations, product quality, features, security measurements, actions to mitigate risks, decision making, pre-market and post-market analysis, continuous monitoring, etc. The European Union has provided guidelines regarding the flow of user data. The common guideline should be listed which manufacturers will follow from the initial stage of product design so that each device can be free from security threats. Moreover, legal organizations from each country or region should develop regulations to protect consumer rights. Legal bodies from different countries will set up laws on how the device and manufacturers will support customers through applications such as monitoring, proving reports, etc.

### 7.7. Education and Training

To avoid attacks due to phishing, social engineering, etc., it is required to allocate a budget from manufacturers or producers to raise awareness among clients and consumers so that they identify the sign of potential attacks due to phishing or reverse engineering. Many users are not quite aware of technological things, so without proper awareness, they could be an entry point for attackers to enter the system. Moreover, technical people need to go through specialized training to maintain the safety of the system and users’ confidential information. A company should develop a friendly user interface where engineers can automatically configure, update, and maintain the network.

## 8. Conclusions

With the emergence and rapid growth of IoT applications, it is drawing continuous attention of attackers and the research community to identify vulnerabilities in security ranging from device attacks to data transport attacks. Moreover, IoE is evolved with the extensive application of IoT in the industry. Furthermore, via the application of intelligent technological innovation, the physical world is being merged with the virtual world, exacerbating the vulnerabilities of IoT-based industrial systems. In this survey, not only are the details of taxonomy of IoT based security and privacy issues discussed, but also countermeasures of each kind of security threat are provided. Different centralized and decentralized security solutions are also discussed. Furthermore, this paper highlighted the IoT ecosystem, applications, functions, and challenges. In the next part, it covered all kinds of security aspects, such as—ABE, ECC, MAC, ML, PUF, and Blockchain for IoT applications based on existing security schemes. In the future, how the aggregation of the quantum system, 5G, FL, AI, and existing centralized and decentralized systems can achieve better data processing, keep data integrity and provide security in the device and data transmission will be explored. Whether Named Data Network (NDN) can achieve better security compared to IP based systems will be discussed. Moreover, different security regulations, ongoing research in the industry, and newly developed attacking methods will also be presented.

## Figures and Tables

**Figure 1 sensors-22-07433-f001:**
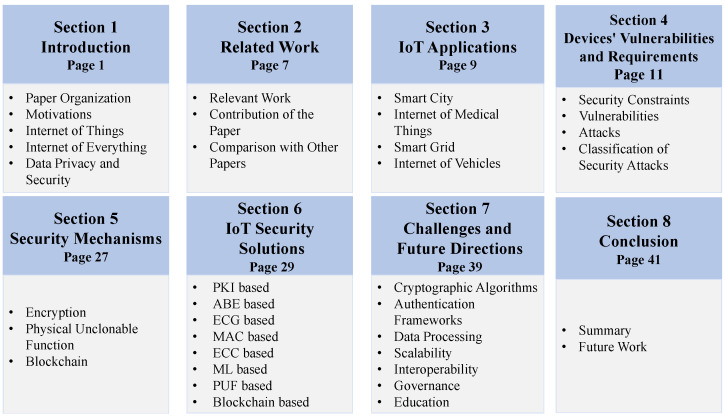
Paper Organization.

**Figure 2 sensors-22-07433-f002:**
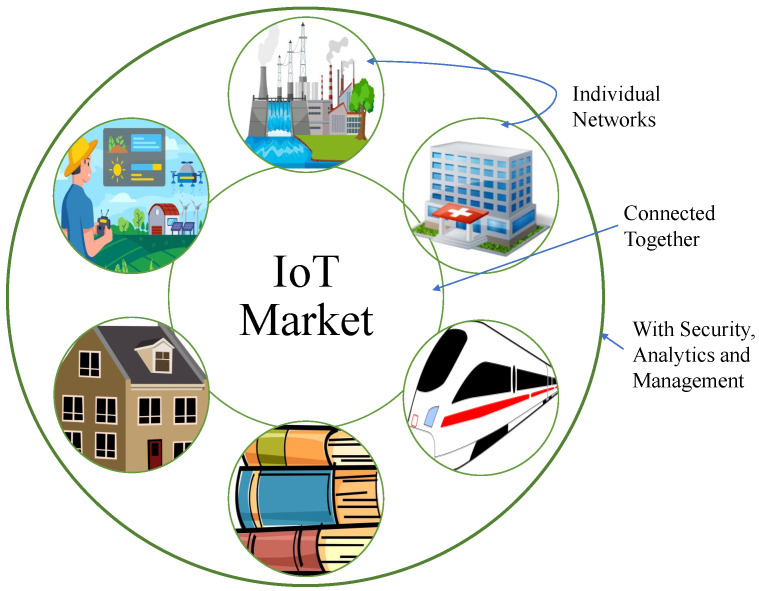
IoT Application.

**Figure 3 sensors-22-07433-f003:**
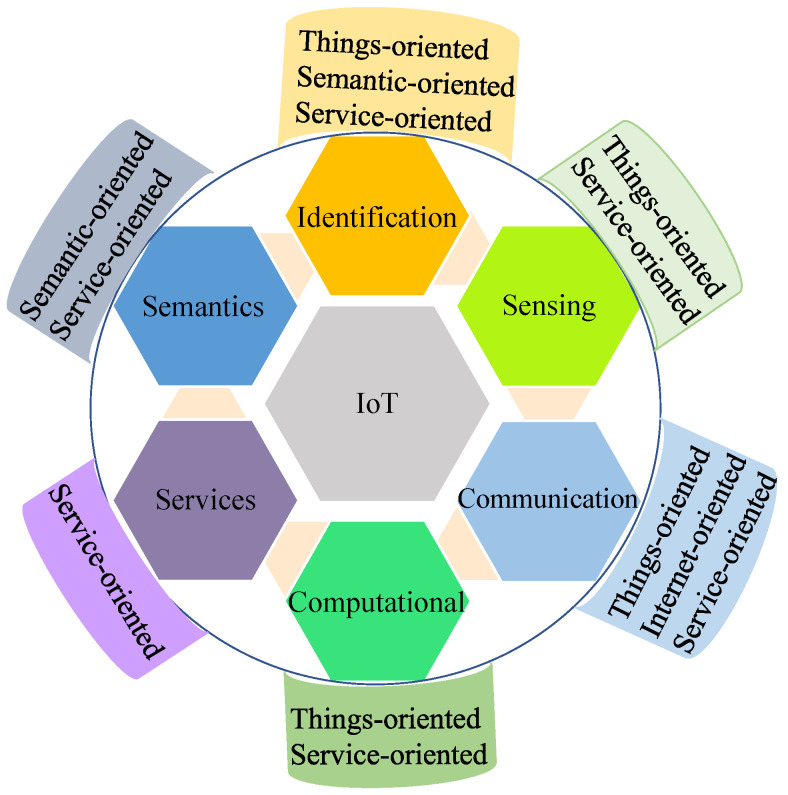
IoT Elements and Paradigm.

**Figure 4 sensors-22-07433-f004:**
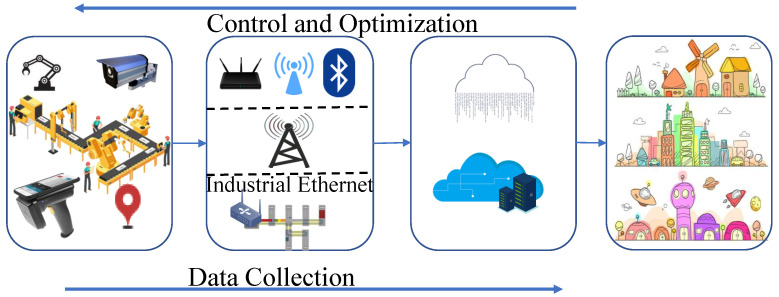
Generalized IoT Architecture.

**Figure 5 sensors-22-07433-f005:**
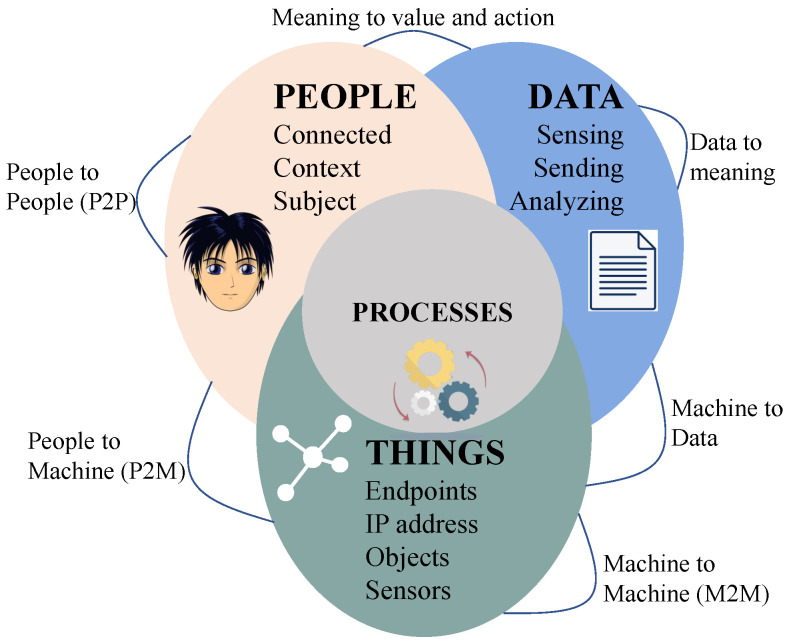
IoE Components.

**Figure 6 sensors-22-07433-f006:**
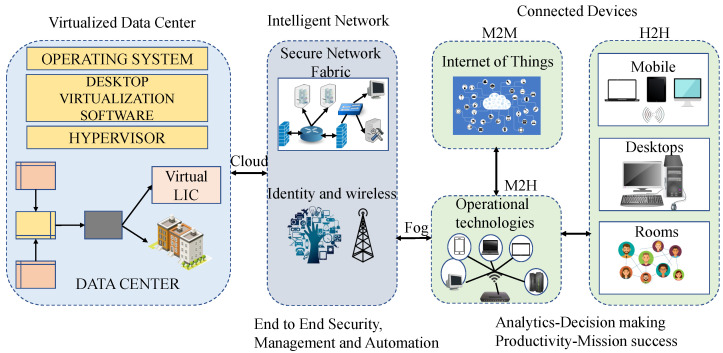
Generalized IoE Architecture.

**Figure 7 sensors-22-07433-f007:**
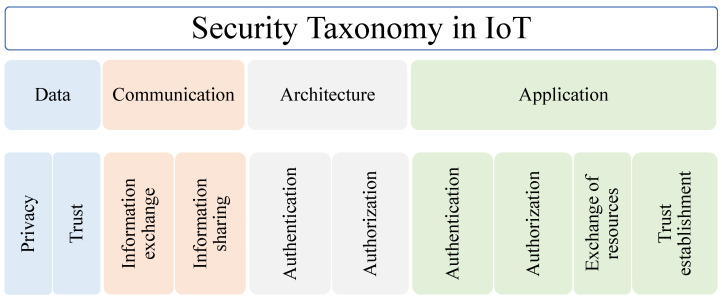
Security Taxonomy.

**Figure 8 sensors-22-07433-f008:**
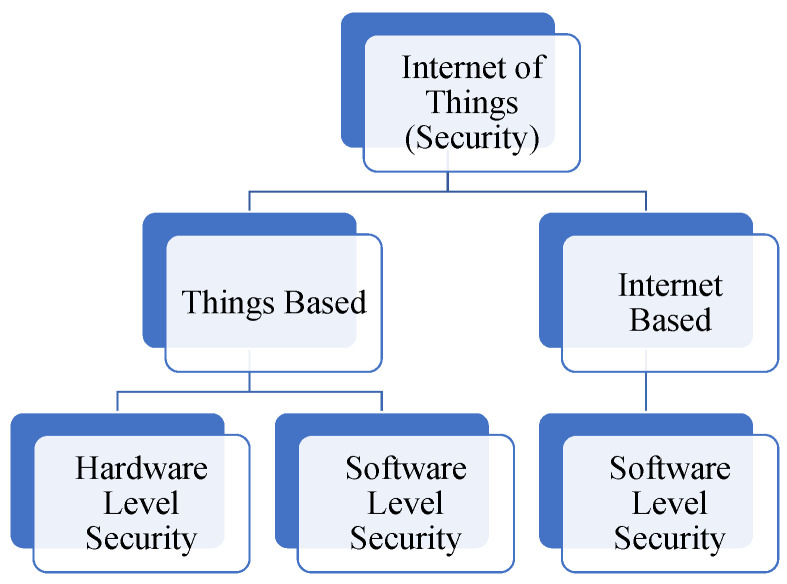
Hardware attack and Software attack.

**Figure 9 sensors-22-07433-f009:**
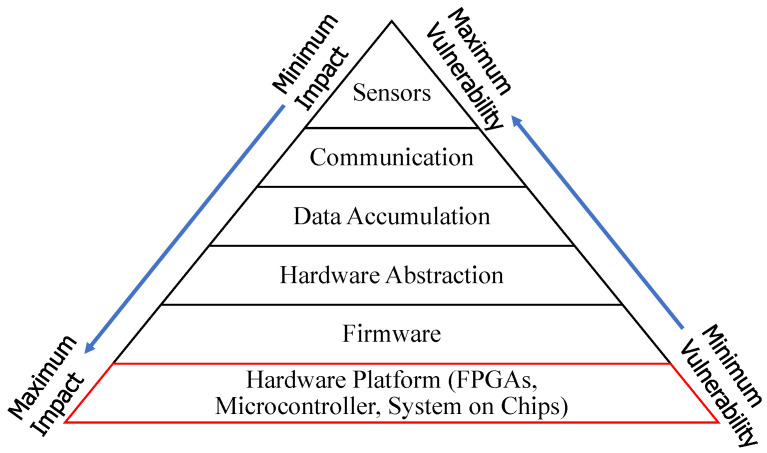
IoT devices attack pyramid based on effects and vulnerabilities.

**Figure 10 sensors-22-07433-f010:**
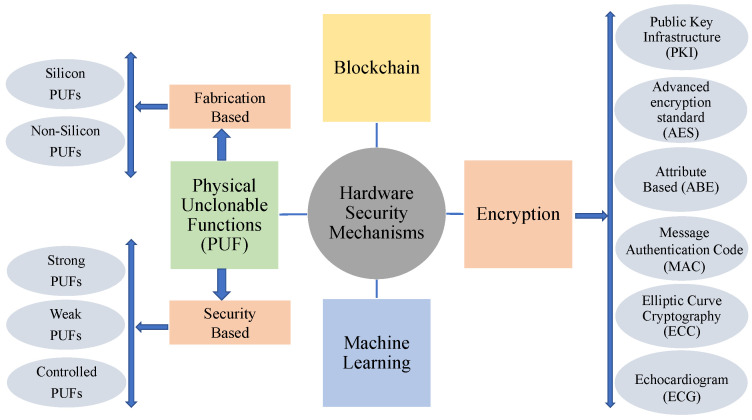
Hardware-based security mechanisms.

**Figure 11 sensors-22-07433-f011:**
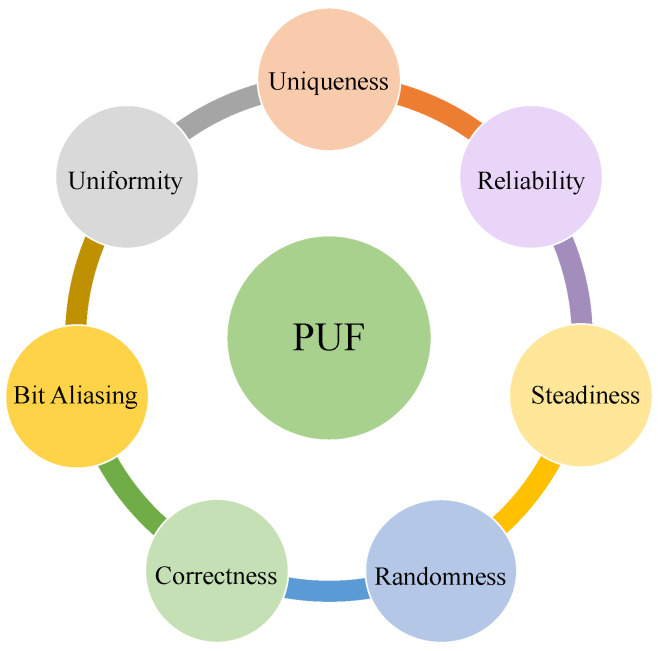
Characteristics of PUF.

**Figure 12 sensors-22-07433-f012:**
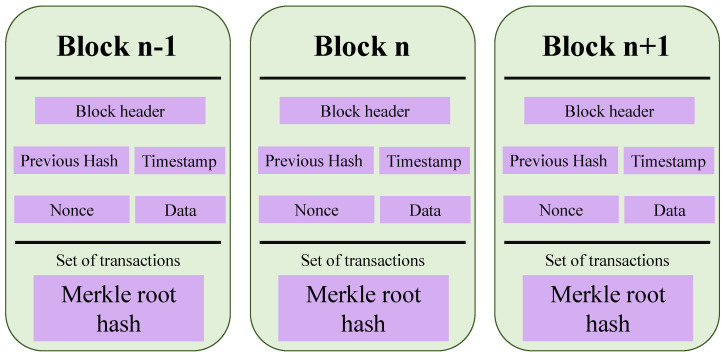
Basic structure of blockchain block.

**Table 1 sensors-22-07433-t001:** Acronyms used in the paper.

Notations	Description	Notations	Description	Notations	Description
IP	Internet Protocol	IoMT	Internet of Medical Things	IoV	Internet of Vehicles
SG	Smart grid	PUF	Physical unclonable function	V2X	Vehicle-to-everything
V2V	Vehicle-to-vehicle	VANET	Vehicular Adhoc Network	V2S	Vehicle-to-sensors
V2I	Vehicle-to-infrastructure	V2N	Vehicle-to-network	V2P	Vehicle-to-pedestrian
RFID	Radio-frequency identification	WiFi	Wireless Fidelity	NFC	Near-field communication
GPS	Global Positioning System	LTE	Long-Term Evolution	MAC	Media Access Control
M2M	Machine to machine	M2H	Machine to human	H2H	Human to human
DoS	Denial-of-service	DDoS	Distributed denial of service	BFA	Brute force attack
PUF	Physical unclonable functions	PKI	Public Key Infrastructure	AES	Advanced encryption standard
ABE	Attribute Based	MAC	Message Authentication Code	ECC	Elliptic Curve Cryptography
RSA	Rivest–Shamir–Adleman	PoW	Proof of work	PoS	Proof-of-Stake

**Table 2 sensors-22-07433-t002:** Comparative analysis of related works.

Survey	Citation	Year	Objective
Understanding security requirements and challenges in Internet of Things	Hameed et al. [30]	2019	Security vulnerabilities are classified; challenges, existing solutions and open research issues are highlighted.
Survey on blockchain for Internet of Things	Wang et al. [40]	2019	Blockchain data structure and consensus protocol enhancement related existing works.
A Comprehensive Survey on Attacks, Security Issues and Blockchain Solutions for IoT and IIoT	Sengupta et al. [11]	2019	Categorize IoT attacks and countermeasures; Detailed IoT and IIoT application-specific blockchain-based solutions.
Demystifying IoT Security: An Exhaustive Survey on IoT Vulnerabilities and a First Empirical Look on Internet-Scale IoT Exploitations	Neshenko et al. [26]	2019	IoT taxonomy, effects and remediation.
IoT: Internet of Threats? A Survey of Practical Security Vulnerabilities in Real IoT Devices	Meneghello et el. [35]	2019	The security measures used by the most prevalent IoT communication protocols, as well as their flaws.
Complementing IoT Services Through Software Defined Networking and Edge Computing: A Comprehensive Survey	Rafique et al. [37]	2020	SDN and edge computing for limited resources computed intensive tasks.
A Survey of Machine and Deep Learning Methods for Internet of Things (IoT) Security	Al-Garadi et al. [27]	2020	IoT security threats and solutions using machine learning and deep learning.
Security, Privacy and Trust for Smart Mobile-Internet of Things (M-IoT): A Survey	Sharma et al. [28]	2020	Threats and countermeasures comparison of existing works for Smart Mobile-Internet of Things.
From Pre-Quantum to Post-Quantum IoT Security: A Survey on Quantum-Resistant Cryptosystems for the Internet of Things	Fernández-Caramés et al. [31]	2020	Comparison of traditional and quantum security vulnerability and impacts.
A Survey on the Internet of Things (IoT) Forensics: Challenges, Approaches, and Open Issues	Stoyanova et al. [32]	2020	Discussed present challenges and solution of IoT forensics.
A Comprehensive Survey on Internet of Things (IoT) Toward 5G Wireless Systems	Chettri et al. [33]	2020	5G layers, effect of 5G on IoT and review on low-power wide-area network for 5G.
Internet of Things for the Future of Smart Agriculture: A Comprehensive Survey of Emerging Technologies	Friha et al. [38]	2021	Emerging technology like SDN, NFV, Blockchain in the smart agriculture related application.
A Survey on the Integration of Blockchain with IoT to Enhance Performance and Eliminate Challenges	Sadawi et al. [41]	2021	Challenges and attacks resistance using blockchain technology.
Applications of the Internet of Things (IoT) in Smart Logistics: A Comprehensive Survey	Song et al. [36]	2021	How IoT can be applied in smart logistics.
A Survey on Security and Privacy Issues in Edge-Computing-Assisted Internet of Things	Alwarafy [39]	2021	Improvement of data processing and resistance of attacks using edge-computing.
Lightweight Cryptographic Protocols for IoT-Constrained Devices: A Survey	Khan et al. [34]	2021	Discussed IoT architecture and lightweight cryptographic protocols.
Public Blockchains for Resource-Constrained IoT Devices—A State-of-the-Art Survey	Khor et al. [42]	2021	Discuss the blockchain technology solutions and how it can be used in resource constraint devices.
Machine Learning-Based Security Solutions for Healthcare: An Overview	Arora et al. [43]	2022	Healthcare security solutions using machine learning.
Security and Privacy Threats for Bluetooth Low Energy in IoT and Wearable Devices: A Comprehensive Survey	Barua et al. [29]	2022	Security threats, taxonomy and solutions for Bluetooth based attack.
A comprehensive survey on machine learning approaches for malware detection in IoT-based enterprise information system	Gaurav et al. [44]	2022	Machine learning based malware detection in the IoT network.
**This Paper**	-	-	Provided overview and comparison of IoT and IoE network; Provided overview of different IoT applications; Details discussion of IoT taxonomy, effects, and countermeasures; Centralized and blockchain based existing security solutions discussion in depth of IoT applications.

**Table 3 sensors-22-07433-t003:** IoT devices security constraints.

Hardware Limitation	Software Limitation	Network Limitation
Computational and energy constraint	Embedded software constraint	Mobility
Memory constraint	Dynamic security patch	Scalability
Tamper resistant packaging		Multiplicity of communication medium
		Multi-Protocol Networking
		Dynamic network topology

**Table 4 sensors-22-07433-t004:** IoT Security Requirements.

Security Requirements	Definition	Abbreviations
Confidentiality	The process in which secret and confidential of the on-air and stored information is strictly preserved and only permitted objects or users have access to it.	C
Integrity	The process in which there is no alternation of data and accuracy is taken care of.	I
Nonrepudiation	The procedure through which an IoT system verifies an event’s legitimacy and origin.	NR
Availability	The process of making sure services are accessible to who needs them, even if there is a power outage or a breakdown.	A
Privacy	The method by which an IoT system is to access private data by following rules and policies.	P
Auditability	The process by which an IoT system monitors it actions.	AU
Accountability	The mechanism through which IoT system users will be responsible for their actions.	AC
Trustworthiness	The method through which an IoT system can verify an individual’s identification and establish trust in a third party.	TW

**Table 5 sensors-22-07433-t005:** Attacks against the various layers of the IoT network.

Physical Attacks	Encryption Attacks	Network Attacks	Application Attacks
Node Jamming	Man In The Middle Attack	Traffic Analysis Attacks	Virus and Worms
Physical Damage	Side Channel Attacks	RFID Spoofing	Malicious Scripts
Node Tampering	Cryptanalysis Attacks	RFID Cloning	Spyware and Adware
Social Engineering		RFID Unauthorized Access	Trojan Horse
Malicious Node Injection		Man In the Middle Attack	Denial of Service
Sleep Deprivation Attack		Denial of Service	Firmware Hijacking
Malicious Code Injection		Sinkhole Attack	Botnet Attack
RF Interference		Routing Information Attack	Brute Force Password Attack
Tag Cloning		Sybil Attack	Phishing Attacks
Eavesdropping		Replay Attack	
Tag Tampering		Hello Flood Attack	
Outage attack		Blackmail Attack	
Object replication		Blackhole Attack	
Hardware Trojan		Wormhole Attack	
		Grayhole Attack	

**Table 6 sensors-22-07433-t006:** Physical attacks with compromised security goals and countermeasures.

Physical Attacks	Compromised Security Requirements	Effects	Countermeasures
Node Jamming	ALL	Communication disruption, Reducing lifetime [99]	Frequency hopping [100], Game theory [99], Spread Spectrum, lower duty cycle, priority messages, region mapping
Physical Damage	ALL	Hardware control, Confidential information leakage [101]	Secure physical design, tamper proof and self-destruction
Node Tampering	ALL	Device ID impersonation [102], Access to sensitive data and Gain access, DoS	Cloning resistance and self-destruction, lowering information leakage (adding randomized delay, deliberately generated noise, balancing hamming weights, strengthening the cache architecture, shielding), integrating Physically Unclonable Function (PUF) in the device [102]
Social Engineering	ALL	Control sensors [103]	Cloud-edge processing and feedback [104], Back up techniques, education of IoT users, tamper proofing and self-destruction
Malicious Node Injection	ALL	Illegal surveillance [102], Control data flow; Man in the Middle	Data compression algorithm [105], Calculation of path credibility [106], Secure firmware update, hash-based mechanisms, Encryption, authentication technique
Sleep Deprivation Attack	I, A, NP	Node shutdown	Intrusion Detection system [63], Firefly algorithm and Hopfield neural network [107], Radial bias function [108]
Malicious Code Injection	ALL	Loss of software integrity, Access to sensitive information and Gain access, DoS	Chain of trust, API endpoint security [102], Traffic monitoring and detection scheme [109], Tamper protection and self-destruction, IDS
RF Interference	ALL	Message block [91], DoS	Distance-based information, Secure kill command for tags, Electronic Product Code (EPC) tags [63], spread-spectrum communication [110], Anti-jamming beamforming scheme [91]
Tag Cloning	ALL	Unauthorized copy of tag	Attack probability scheme [111], Tag randomization [112], Encryption, hash-based methods, authentication framework, kill sleep instruction, isolation, blocking, distance estimation, Integrating PUFs into RFID tags
Eavesdropping	C, NR, P	Extract critical network information [102]	Secure Bootstrapping [63], low-cost demilitarized zone [102], Encryption techniques, shift data to the back end
Tag Tampering	ALL	Malicious altercation of data in tag memory [113]	Authentication watermark and recovery Watermark [114], Integration of PUFs into RFID tags, hash-based mechanisms, encryption, tamper-release layer RFID, integrating alarm option for active Tags
Outage attack	A, AC, P, AU, NP	Disrupt or bias the state of applications [115]	Random time hopping sequence and random permutation [115], Secure physical design
Object replication	ALL	Control network [116]	3-D backward key chains based on deployment knowledge [117], Encryption, Hash-based methods, Lightweight cartographic schemes
Hardware Trojan	ALL	Function change of chips and sensitive information leakage [118]	Temporal thermal information [118], Electromagnetic radiation [119], Side-channel signal assessment (based on path-delay fingerprint, based on symmetry breaking, based on thermal and power, machine learning application), trojan activation

**Table 7 sensors-22-07433-t007:** Encryption attacks with compromised security goals and countermeasures.

Encryption Attacks	Compromised Security Requirements	Effects	Countermeasures
Man In The Middle Attack	C, I, P, NR	Extract critical network information [102]	Encryption of the RFID communication medium, authentication techniques [96]
Side-Channel Attacks	C, AU, NR, P	Secret key [84]	Shielding [122], Adding randomized delay [123], Randomization [124], Blocking, isolation, sleep instruction, kill command, tamper proofing and self-destruction, lowering information leakage, obfuscating methods
Cryptanalysis Attacks	ALL	Encryption Key	Ultra-lightweight cryptography algorithm (SLIM) [125]

**Table 8 sensors-22-07433-t008:** Network attacks with compromised security goals and countermeasures.

Network Attacks	Compromised Security Requirements	Effects	Countermeasures
Traffic Analysis Attacks	P, NR	Data leakage (Network information) [126]	Anonymous Communication Scheme Based on the Proxy Source Node and the Shortest Path Routing [147], Traffic flow in different gateway [126]
RFID Cloning	ALL	Genuine tag removal and data collision [148]	Firewalls, encryption of RFID signals, authentication to identify approved users, and shielding of RFID tags, PUF [96], Unreconcilable collision detection by multiple tags of same ID [148]
RFID Spoofing	ALL	Data Manipulation and Modification (Read, Write, Delete)	RFID authentication protocol, Data encryption [96], Elliptic Curve Cryptography [149],
RFID Unauthorized Access	ALL	Data manipulation [97], Data Modification (Read, Write, Delete)	Authentication, Field detectors, Shift data to the backend [96], Arbitrator over the Universal Software Radio Peripheral (USRP) platform [150], Control circuit to notify and software [151]
Man In the Middle Attack	C, I, P, NR	Data Privacy violation, source integrity [152]	Encryption, Secure channel, Authentication protocol, Multidimensional plausibility check [132], Radio frequency fingerprint technology [153]
Denial of Service	P, A	Network Flooding, Network Crash	One-way hash chain [154], Policies provided by providers
Sinkhole Attack	A, C, I	Routing protocol [155], Data alteration or leakage	Watchdog scheme, pathrater scheme [156], Geographic routing protocol [155], Radial bias function [108], Attribute-based access control and trust-based behavioral tracking [157], IDS solution, parent fail-over, identity certificates, and a rank authentication scheme
Routing Information Attacks	I, NR	Routing Loops	Authentication, localized encryption and authentication protocol (LEAP) [156], Sink-based intrusion detection system [158]
Sybil Attack	C, I, AC, NR, P	Multi-path routing [156], Unfair resource allocation, Redundancy	Radio resource testing and random key predistribution [156], Rule-based anomaly detection system [159], ID-based public keys [160], Attribute-based access control and trust-based behavioral tracking [157], Artificial bee colony (ABC) of honey bees [139], Classification-based Sybil detection (BCSD)
Replay Attack	ALL	Network congestion, DoS	Key agreement based on ECC, PUF, Hash etc [161], Gaussian-tag-embedded physical-layer authentication (GTEA) framework using a weighted fractional Fourier transform (WFRFT) [162], A challenge and response technique, the time-based or counter-based mechanism
HELLO flood attack	C, I, AC, NR, P, A	Advertise fake route, Routing protocol [156], Resource consumption [163]	Link-layer metric as a feature in the determination of the default path, limit size of connections [110], Amalgamation the merits of probability-based and location-based flooding algorithms [164], Deep learning [163]
Blackmail attack	C, I, P, NR	Isolation of legitimate device [143]	Deep learning [165], Authentication mechanism
Blackhole attack	C, I, AC, NR, P	Drop packets [156]	Multiple routing path [110], Attribute-based access management and trust-based behavioral tracking [157], Ad-hoc on-demand distance vector (AODV) with encryption [166], Proactive reputation updating algorithm and a reputation-based probabilistic forwarding strategy [167]
Wormhole attack	C, I, AC, NR, P	Routing protocol [155], Delete files and documents [84], Packet tunneling	packet leash [156], Markle tree authentication, binding geographic information, Geographic routing protocol [155], Increment of alternative routing path and reduce timeout duration [144]
Grayhole attack	C, I, AC, NR, P	Packet drop [145]	Multiple routing path [110], Acknowledgment count [168], Overhear the transmission of neighbor node for failure detection framework [169], Multihop communication [145]

**Table 9 sensors-22-07433-t009:** Application attacks with compromised security goals and countermeasures.

Application Attacks	Compromised Security Requirements	Effects	Countermeasures
Virus and Worms	ALL	Resource Destruction	Machine learning techniques and signature matching techniques [186], Security updates, side-channel analysis, verify software integrity, control flow, protective Software
Malicious Scripts	ALL	Infected Data	Algebra statistics and geometric trends [187], Firewalls
Spyware and Adware	ALL	Resource Destruction	Installing antivirus or antimalware; API call network with the heuristic detection mechanism [188], Logging and testing [173]
Trojan Horse	ALL	Resource Destruction	Noise bound [189], Monitoring, Audit and research device and network, place IoT devices in separate segment or virtual LAN [190]
Denial of Service	A, AC, AU, NR, P	Network shutdown, Resource exhaustion [64]	Access Control Lists
Firmware Hijacking	ALL	Unauthorized operation and path for other attacks [180]	Electromagnetic pattern [191], use Safe programming languages, add runtime code, audit software
Botnet Attack	ALL	Service outage [181]	Packet monitoring and feed to Enhanced Support Vector Neural Network (ESVNN) [192], Deep learning [193], Bot program in local DNS server [181]
Brute Force Password Attack	ALL	Sensitive data, System compromise [194]	Logging attempt and frequency [195], Dynamic password change [196], Securing firmware update, cryptography methods
Phishing Attacks	ALL	Stage for multiple attack [197]	Machine learning [197], Artificial intelligence [198], Cryptographic methods

**Table 10 sensors-22-07433-t010:** Comparative analysis of IoT security schemes.

Author	Year	Objective	Technique Used	Type of Data	Framework	Pros	Cons
Guo et al. [214]	2017	Solve complexity due to dynamic data	PKI	Big data	Big data left	Single sign-on	ID exposed, Fake certificate after changing RSU
Li et al. [215]	2019	Remove certificate storage dependency	PKI	Vehicle data	-	Certificate-less	vulnerable to resist attack
Kerrache et al. [218]	2019	To ensure trust among drivers	Chaotic map and Social Network	Social profile	TACASHI	Honesty factor	Dependent on external factor
Meshram et al. [221]	2021	To secure smart city	Chaotic map	Smart city	-	Lightweight	Need to enhance security
Al et al. [219]	2021	Reduce communication and key management overhead	CRSA	Throughput gain	-	Minimizes collision and Improved throughput	High storage, Keys of vehicles can be exposed
Hwang et al. [223]	2020	To enable safe channel for sharing medical data	CP-ABE	Medical data	-	Safeguard issue of key abuse	Leakage of PHI file
Han et al. [222]	2021	Algorithm to build frequent item sets	CP-ABE	Attributes	-	Improved speed	Limited to same attribute set
Huang et al. [224]	2019	To safeguard against unauthorized entity	ECG	PHR file	-	Lightweight	Need to be anonymous
Hahn et al. [228]	2019	Countermeasure against MAC-based flaws	MAC	Health data	-	Less verification time	Server impersonation
Siddiqi et al. [227]	2020	To build feasible and secure IoMT communication	MAC	Medical data	IMDfence	Less energy consumption	Need to ensure anonymity
Wazid et al. [229]	2019	Secure wireless communication	ECC	Surrounding information	AKM-IoV	Dynamic node addition	Does not resist DoS attack
Wu et al. [230]	2020	Reduce the verification delay and achieve fast message verification	ECC	Batch message	-	Batch verification	Need focus on security
Thumbur et al. [231]	2021	Avoid complex certificate management problem and key escrow problem	ECC	Signature	Aggregate Signature	Low verification time and storage at RSU	DoS attack
Zhang et al. [232]	2020	Secure communication with limited bandwidth	ECC	Signature	-	Random number to avoid side-channel attack	Need RSU and TA secure communication
Ghahramani et al. [233]	2020	Support roaming users in global mobility network	ECC	Mobile users data	-	Added deep learning to verify biometric	High storage and communication cost
Xie et al. [235]	2021	To protect wireless sensor networks in smart city	ECC	Wireless sensor	-	Simple	Smart card stolen attack
Xia et al. [237]	2021	To secure environment of smart grid	ECC	Smart meter	-	Completed in two steps	Smart meter capture attack
Chen et al. [238]	2021	To get data from edge nodes	ECC	Edge utility nodes	-	XOR and Hashed computation	Did not consider the whole network
Srinivas et al. [239]	2021	Secure big data collection in smart transportation	ECC	Vehicle data	-	Security	Computational, communication cost
Sharma et al. [243]	2021	Correctness of data exchange	Supervised machine learning	Location and movement	-	Detect attack and countermeasure	Limited to position based attack
Pascale et al. [244]	2021	Detection of a possible cyber-attack	Machine learning	Parameters as RPM	-	Accuracy	Not focused on data transmission
Pradeep et al. [245]	2022	Secure smart city applications	CHAP PPP	Operational data of city	-	Simple calculations	System verification is absent
Wang et al. [246]	2018	Physical and external security	PUF	Vehicle data	NOTSA	OBU with segregated applications	Need secure area in OBU
Yanambaka et al. [250]	2019	To develop secure IoMT system	PUF	Health data	PMSec	Simple and lightweight	ML attack, MITM attack
Alladi et al. [247]	2021	Verification of ECU firmware	PUF and ECC	Firmware and software	-	Physical safety	Insider can identify secret key
Aman et al. [248]	2021	Resist physical attacks and reduce overhead with secure communication	PUF	Network traffic	-	Low authentication packets and overhead	CRP updates can expose system
Alladi et al. [249]	2021	Sensitive information transfer and resist node capture/tampering attack	PUF	Traffic information	SecAuthUAV	Low energy in device and low storage in server	Could server impersonation and ID expose
Badar et al. [251]	2021	Securing smart grid	PUF	Line flaw or breakage	-	Computational cost is low	Communication cost is high
Tanveer et al. [252]	2021	To make reliable smart grids	PUF and ECC	Power usages	ARAP-SG	Computational cost is less	Unnecessary storage and computation
Jiang et al. [254]	2019	Application of blockchain in IoV	Blockchain	Big data	-	Showed IoV using Blockchain	Security will be in future work
Liu et al. [255]	2019	Reduce transaction confirmation delay and clod-start of new users	Blockchain (PoW)	Traffic information	-	Pricing strategy	Not suitable for high resource adversary
Yin et al. [256]	2020	Reduce processing time with gainig profit	Blockchain	Mobile crowdsensing	-	Time-window based urgent task	Reactive security
Yang et al. [257]	2019	Trust management in vehicular network	Blockchain (PoW and PoS)	Traffic	-	Credible neighbor rating	Reply attack, MITM attack etc, overhead
Gao et al. [258]	2020	Effective network management	Blockchain	Vehicular data	SDN	Avoid frequent handover and relieve pressure	No focus on data transmission trust
Xu et al. [259]	2021	Energy efficiency and encounter external invasion	Blockchain	Vehicular data	-	Suitable for high amount of data	basic security
Javaid et al. [260]	2020	Trust establishment	Blockchain (dPoW) and PUF	Traffic	-	No physical and side-channel attack	Can expose response
Vivekanandan et al. [261]	2021	Secure device to device communication in smart city	Blockchain and ECC	Devices data	BIDAPSCA5G	Location incorporation	Eavesdropping attack
Wang et al. [262]	2022	To build a reliable communication channel for healthcare	Blockchain (PoW) and PUF	Health information	-	Low cost	Storage cost

## Data Availability

Not applicable.

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
