# Peer review of "Internet of Things: Security and Solutions Survey"

_sensors, 2022, doi:10.3390/s22197433_

Round 1

Reviewer 1 Report (New Reviewer)

Summary:
The manuscript is a survey paper it is aimed to assist researchers by classifying attacks/vulnerabilities based on objects. The method of attacks and relevant countermeasures are provided for each kind of attacks in the document presented.

In the manuscript presented, the authors address a highly interesting and very current topic. In addiction, the manuscript is well organized and structured, nevertheless I consider that the authors should attend to certain minor  issues in order to improve the quality of work.

Comments and Suggestions for Authors:

- Fig. 1 does not show the number of section according to numbers mentioned in the text.  (1.2 Paper Organization).

- Table 1 shows V2V acronym as Vehicle-to-everything and the right acronym to Vehicle-to-everything is V2X -- V2V is for Vehicle-to-Vehicle (communication).

- Table 1 does not show some acronyms used in the document as H2H, V2V, etc.

- Authors should modify figure 17 by increasing the size of the arrows because they are very small.

- Even if the reader knows the subject, the authors should expand the first occurrence of any the acronyms or initialism, some of them are not expanded as examples RFID, IC, SoC, VLSI, GPS, H2H, SCADA, etc.

- References to standards, de facto standard, services, equipment and components must be included, as example Wi-Fi, ZigBee, etc.

- Page 19, there a link (https://wiki.owasp.org/index.php/ASP_Internet_of_Things_Project#tab=IoT_Attack_Surface_Areas) that should be move to reference section and to assign reference number.

- Page 22, there a link (https://wiki.owasp.org/index.php/OWASP_Internet_of_Things_Project#tab=IoT_Vulnerabilities) that should be move to reference section and to assign reference number.

- Authors should review the use of references due to some references are not used.

- Some typos should be corrected, as examples: page 23 appears Propoer instead of Proper.

Author Response

Dear Reviewer,

Hope you are doing well. We have edited the manuscript and please see the attachment where we have explained the edits.

Reviewer 2 Report (New Reviewer)

In this paper authors introduced a survey aimed to classify security attacks, vulnerabilities, countermeasures as well as suggestions to future research directions. The topic is very important and sound, however, several concerns are summarized in the following points:

- English language should be edited in terms of style and typos. i.e authors repeated writing "survey paper" more than three times only in the abstract. It is of course known this is a paper, so it is enough to write "survey".

- The aim and goal of the paper is not concise and consistent in the whole paper. In some places it is mentioned that this paper is aimed to survey security issues in IoT and in other places additional platforms are also combined , i.e, IoE and IoMT as in the contribution of the paper in page 7.

- So many basic information is provided. However, such information are essential and  any reader who has basic knowledge about IoT should be aware of, i.e, the unnecessary introductions to the basic topics such as IoT, etc.

- Authors discussed the challenges of Smart City Development. I could not find any relation of this topic  with IoT security issue. What is the purpose of such summary? Same comment for IoMT, global market!

- As a summary , sections 2 and 3 are basic information and has no relation with the survey subject. Authors may introduce each application area in one paragraph and highlight the security related issues only. This is enough for the purpose of this survey.

- Pages 19, 22: it is not proper to include links in the body of the paper, it can be referenced as well.

- The details of OWASP system are not necessary in the perspective of this survey.

- Figures in the whole paper should be referenced if they are copied or adopted from other resources.

- The citation of related works in subsection titles is not recommended. For example: in page 22 the subsection title is like this: “4.3.1. Weakness of Authentication System [107] [108] [109] [110] “. Instead, authors should cite the works in-text and refer to them in  conclusion or individually. The same comment for many other subsections in the paper.

-Page 24: The demonstration of security requirements is very basic and such information are known to the beginners in information security.

- According to this reviewer's knowledge, the protocol stack of IoT is different from that public protocol TCP/IP as IoT protocols are application dependent. Therefore Table 5 shows information that is not useful in the context of IoT. The layers of IoT infrastructure are even different from the standard layers of TCP/IP.

- Finally, The most important part of the paper which is the future directions in a survey paper, should be concluded in light of the works discussed in the paper and should be connected to the problem of existing literature. However, most of the issues in section 7 are general and are somehow clear in the IoT security such as lightweight encryption and lightweight security models. What is needed in this section is an in-depth reflection of the issues and unsolved research gaps stated in the body of the paper in sections 5 and 6.

Author Response

Dear Reviewer,

Hope you are doing well. We have edited the manuscript and please see the attachment where we have explained the edits.

Reviewer 3 Report (New Reviewer)

In this paper authors provide a comprehensive survey of security threats faced by IoTs and the solutions proposed by various researchers. This is one of the most comprehensive surveys on IoTs that I have come across in past. The structure of the paper is well organized. The organization of the paper is well described and easy to follow. The authors start with the IoT applications, then they discuss their security vulnerabilities, the proposed countermeasure, and also discuss in detail the challenges and future directions. Since, it is a survey paper, so no groundbreaking novelty is expected from authors anyway. But the way they have covered state-of-art is quite nice. I have few comments on the paper though that should be addressed before the acceptance of the paper.

1. The abstract of the paper needs to be rephrased. The quality of English is not up to the standard. For example, the word 'also' was used quite a few times in last couple of sentences of abstract. 

2. Apart from the abstract, the spelling and grammatical mistakes can be noticed across the paper and the paper needs a thorough review in this regard.

3. The citations of the paper are not in chronological order which does not give a good impression. For example, second citation in introduction section is numbered [7].

4. The introduction section was quite lengthy and sometimes it became too wordy and became hard to follow. If the authors could break it down into further subsections or somehow maintain the fluidity in the paper, them it would be great.

5. Contribution in Tables 2 is summarized in a good way. But my question remains that how you are different? What you offer that others do not. This should be the real contribution. Just presenting a Table is not enough. A discussion on that is also required.  

6. Heading of section 4.3.3 is not correct and similar mistakes (spelling or grammatical) can be seen throughout the paper.

Apart from the comments given above, the paper seems to be OK and can be considered for publication after revision.

Author Response

Dear Reviewer,

Hope you are doing well. We have edited the manuscript and please see the attachment where we have explained the edits.

Round 2

Reviewer 2 Report (New Reviewer)

Authors should further address the comments that are not being addressed in this revision:

1-Authors discussed the challenges of Smart City Development. I could not find any relation of this topic with IoT security issue. What is the purpose of such summary? Same comment for IoMT, global market!

2- As a summary , sections 2 and 3 are basic information and has no relation with the survey subject. Authors may introduce each application area in one paragraph and highlight the security related issues only. This is enough for the purpose of this survey.

3- The details of OWASP system are not necessary in the perspective of this survey. (Information from wikipedia should be avoided).

Author Response

Dear Reviewer,

We have modified the manuscript as per your valuable suggestions. Please have the attachment.

This manuscript is a resubmission of an earlier submission. The following is a list of the peer review reports and author responses from that submission.

Round 1

Reviewer 1 Report

This is a very comprehensive review of a large number of security threats and potential solutions, but I have the following main issues:

- It is not clear to me which security threats are specific to IoT and which are not. This is not discussed in the paper in a clear manner, given that the scope of the work is IoT, and almost all of the threats, vulnerabilities, solutions etc. are applicable to most of the information security landscape, not just IoT.

- A security taxonomy of IoT would have been a good contribution of this paper but unfortunately the one produced here lacks important details like its motivation, the methodology and a logical structure.

- At 60 pages, this work is too long. The recommended length of 20 pages would have been better.

Author Response

Dear Professor,

Thank you for your valuable feedback which helps us to improve the manuscript. Hope our modification of the manuscript will satisfy your thoughts.

Regards,

Authors of the manuscript

Reviewer 2 Report

The article provides a comprehensive overview of IoT devices and their security solutions. It is widely known that IoT devices are vulnerable in security, so research in this area is both warranted and desired.

Here are some remarks:

1. The article is extremely long which may discourage the reader.
2. The article contains too many things that are already common knowledge  chapter 4.1 is unnecessary to elaborate on security threats, a short table would be enough.
3. Some sentences are too long to follow, so it is recommended to break them into short but meaningful ones to make the manuscript readable.
4. Quite a lot of abbreviations are used in the manuscript. To improve readability, it is recommended to use a table to locate all frequently used abbreviations with their descriptions.
5. I suggest to increase the number of studies and add a new discussion there to show the advantage. Add DRDoS attacks and assess and compare what the behavior would be in an industrial setting.
6.Maybe consider the following studies that clearly demonstrate the vulnerability of DRDoS attacks in industry:
7.Mitigation against DDoS Attacks on an IoT-Based Production Line Using Machine Learning
8.The vulnerability of securing IoT production lines and their network components in the Industry 4.0 concept
9.DDoS reflection attack based on IoT: A case study
10. Resources of the nature of : https://wiki.owasp.org/ are not suitable for this type of work.

Author Response

(The authors gave the same response as above.)

Reviewer 3 Report

This paper presents a really extensive survey on the security of Internet of Things (IoT) and Internet of Everything (IoE). The authors provide a long list of references on this topic, detailing the aim of these and the reader can easily get many useful references concerning different aspects of security in IoT, such as different attacks and known vulnerabilities and some countermeasures which include cryptographic solutions in terms on encryption and authentication schemes and protocols. 

However, although the presentation, different explanations and list of references is quite adequate, this reviewer thinks that some aspects could be enhanced. 

Firstly, there is a huge collection of acronym and the non-expert reader (one of the possible targets of this paper) could get lost and this reviewer suggests to introduce the meaning of all them, including a reference (if possible for them). This includes, for instance, all cited cryptographic schemes, such as ECC or RSA. This would help the reader to locate the original work rather than a paper where to find the corresponding scheme as a reference or where it is not clearly or incorrectly explained, as is the case of reference [239], where commutative RSA is introduced, which is not really a crypto system, but also the use of a property of such crypto system, namely the commutativity of the encryption and decryption processes with two different pairs of public-private keys (which is not an exclusive property of RSA). 

Secondly, this reviewer suggests to increase the citations to ECC based security schemes. ECC has a big importance in networks formed by light devices due to its low requirements and this is shown by the big amount of authors existing in the literature that use this solution. As an example, the reference to RSA can be found in 22 pages, whereas to ECC appear in just 13. 

As a third comment, this reviewer encounters that the authors considered mainly one-to-many and many-to-one communications and there exists many examples of many-to-many communications. In this setting arises a main problem, which is authentication due to the lack of a Public Key Infrastructure (PKI). Thus some alternative solutions has to be found, such as the use of identity based protocols or even the more recently and extended blockchain technology. 

Another problem that arises in group communications is Group Key Management. All the references that the authors are including consider a classical one-to-one communication to a key server, which increases the amount of keys to be stored as well as the bandwidth used. There exist one-to-many solutions for key distribution and even many-to-many key management schemes, which, in the latter case, the group key is built in a collaborative way, and that provide solutions to the former technical problems.

Finally, this reviewer does not agree in the following assertion appearing in the first paragraph of Section 5: “The invention of quantum computers will make software-based security solutions vulnerable.”, which is true just in part. Quantum computers will menace software and hardware solutions based in the most extended cryptographic protocols such as those based in the integer factorization and the discrete logarithm problem, as it is remarked in the title of the original Peter Shor’s paper, which includes, from RSA cryptosystem, to applications of ECC, such as blockchain. NIST is nowadays analyzing different proposals for the post-quantum era and its results should be a new standard to be used both for software and hardware solutions. Thus, what is asserted just following the authors’ above statement is not true at all, I.e., “Therefore, the hardware-based solution is one of the possible solutions due to the risk factor of existing software-based security.”

By the preceding this reviewer recommends a major revision of the manuscript previous to a different recommendation for its possible publication. 

Author Response

(The authors gave the same response as above.)

Round 2

Reviewer 1 Report

My original comments still hold; in my opinion the authors have not made significant changes to the paper to warrent its acceptance.

Reviewer 2 Report

The authors had addressed all my concerns and I have no more comments.

Reviewer 3 Report

The new version of the paper is very well written and organized and fulfills this reviewer suggestions. Under this reviewer's opinion, the paper is worthy of publication.